# Allosteric inhibition of a stem cell RNA-binding protein by an intermediary metabolite

**Carina C Clingman, Laura M Deveau, Samantha A Hay, Ryan M Genga, Shivender MD Shandilya, Francesca Massi, Sean P Ryder\***

Department of Biochemistry and Molecular Pharmacology, University of Massachusetts Medical School, Worcester, United States

**Abstract** Gene expression and metabolism are coupled at numerous levels. Cells must sense and respond to nutrients in their environment, and specialized cells must synthesize metabolic products required for their function. Pluripotent stem cells have the ability to differentiate into a wide variety of specialized cells. How metabolic state contributes to stem cell differentiation is not understood. In this study, we show that RNA-binding by the stem cell translation regulator Musashi-1 (MSI1) is allosterically inhibited by 18–22 carbon ω-9 monounsaturated fatty acids. The fatty acid binds to the N-terminal RNA Recognition Motif (RRM) and induces a conformational change that prevents RNA association. Musashi proteins are critical for development of the brain, blood, and epithelium. We identify stearoyl-CoA desaturase-1 as a MSI1 target, revealing a feedback loop between ω-9 fatty acid biosynthesis and MSI1 activity. We propose that other RRM proteins could act as metabolite sensors to couple gene expression changes to physiological state.

\*For correspondence: sean. ryder@umassmed.edu

**Competing interests:** The authors declare that no competing interests exist.

**Reviewing editor**: Douglas L Black, Howard Hughes Medical Institute, University of California, Los Angeles, United States

## Introduction

The RNA-binding protein Musashi-1 (MSI1) is expressed in stem and progenitor cells of neural and epithelial lineage. In the central nervous system, MSI1 is expressed in astrocytes and committed glial and neural progenitor cells, but not in mature neurons and oligodendrocytes (*Figure 1A*; *Kaneko et al., 2000*; *Dobson et al., 2008*). In histological studies of neural and epithelial tissues, MSI1 is routinely used as a marker for stem and progenitor cells (*Johnson et al., 2010*). Analysis in mice and primary cells shows that MSI1 regulates neural development. *Msi1⁻ᐟ⁻* knockout mice are uncoordinated, ataxic, develop hydrocephaly, and die within 1–2 months after birth (*Sakakibara et al., 2002*). Their brains are small, contain an expansion of early lineage progenitor cells, and display fewer mature cell types than normal (*Sakakibara et al., 2002*). Embryonic neurospheres cultured from *Msi1⁻ᐟ⁻* mouse brains have a reduced capacity to differentiate into mature neurons and oligodendrocytes (*Sakakibara et al., 2002*). In primary oligodendrocyte progenitor cells, MSI1 promotes progenitor cell survival and prevents differentiation into mature oligodendrocytes (*Dobson et al., 2008*). The phenotype and expression pattern reveal that MSI1 plays an early role in regulating neurogenesis and gliogenesis.

MSI1 contains two RNA recognition Motifs (RRMs) and is homologous to *Drosophila melanogaster* Musashi, a post-transcriptional regulatory protein that guides external sensory bristle patterning in flies (*Sakakibara et al., 1996*). In vitro SELEX experiments identified a series of aptamer sequences that bind to MSI1 (*Imai et al., 2001*). Visual inspection identified a consensus sequence (G/A)U₁₋₃AGU that was present in most but not all of the aptamers. A number of MSI1 targets have been identified by co-immunoprecipitation, including NUMB, a repressor of NOTCH signaling. *Numb* transcripts harbor MSI1 consensus elements in the 3'-UTR (*Imai et al., 2001*). MSI1 interacts with the *Numb* 3'-UTR in vitro, and *Numb* mRNA co-immunoprecipitates with MSI1 in transiently transfected NIH 3T3

**eLife digest** When an embryo is developing, stem cells must divide and develop into many specialized types of cells. However, if cell division doesn't stop, or if it restarts later in life, it can cause tumors to form.

Musashi-1 is a protein that binds to molecules of RNA and helps to promote cell growth during development: mice that lack this protein have serious brain defects and die shortly after birth. Musashi-1 is usually turned off in adult cells that are not dividing. Sometimes, however, it remains active and contributes to the growth of cancers in the brain and the gut. Reducing Musashi-1 levels in colon tumors slows their growth and causes the cancer cells to die.

To find a compound that would switch off Musashi-1, Clingman et al. screened more than 30,000 compounds and identified four inhibitors. One of these was oleic acid, a fatty acid that is found in olive oil and other animal and plant oils. Oleic acid interacts with Musashi-1 in a way that changes the shape of the protein. These changes mean that Musashi-1 is no longer able to regulate the genes that control cell proliferation.

Clingman et al. also found that Musashi-1 promotes the activity of a particular enzyme that makes fatty acids; molecules that are needed in large quantities when cells are dividing. Musashi-1 appears to act as a 'nutrient sensor', turning down the activity of this enzyme in cells when levels of oleic acid are high, and turning up enzyme activity when oleic acid levels are low. The findings of Clingman et al. further reveal how our diets can affect gene expression, and may aid the development of new treatments against cancer.

cells. Overexpression of MSI1 in NIH 3T3 cells decreases NUMB protein levels without affecting *Numb* mRNA and reduces the expression of a luciferase reporter in a 3'-UTR dependent manner (*Imai et al., 2001*). Together, the results show that MSI1 negatively regulates *Numb* mRNA translation. In contrast, MSI1 acts as a translational activator in *Xenopus laevis* oocytes, where it modulates cell cycle progression by regulating mRNA encoding the proto-oncogene Mos (*Charlesworth et al., 2006*).

MSI1 also promotes proliferation of numerous cancers of the brain and epithelial tissues (*Toda et al., 2001*; *Hemmati et al., 2003*; *Yokota et al., 2004*; *Sanchez-Diaz et al., 2008*; *Sureban et al., 2008*). MSI1 depletion in medulloblastoma and colorectal tumors results in decreased proliferation and increased apoptosis (*Sanchez-Diaz et al., 2008*; *Sureban et al., 2008*). In colorectal tumors, MSI1 depletion is accompanied by inhibition of Notch-1 and upregulation of p21$^{WAF1}$, a MSI1 target involved in cell cycle regulation (*Battelli et al., 2006*; *Sureban et al., 2008*). Musashi-2 (MSI2) is 69% identical to MSI1 protein and is expressed in a partially overlapping set of tissues (*Sakakibara et al., 2002*). MSI2 regulates hematopoesis and is involved in acute myeloid leukemia (*Ito et al., 2010*; *Kharas et al., 2010*). In myeloid leukemia cells, MSI2 is highly expressed, and depletion results in decreased proliferation and increased apoptosis (*Kharas et al., 2010*). The crisis phase of myeloid leukemia is marked by low NUMB expression (*Ito et al., 2010*). Loss of MSI2 restores NUMB expression and impairs the blast crisis phase of myeloid leukemia (*Ito et al., 2010*). Ultimately, MSI2 expression levels are directly correlated with poor prognosis in myeloid leukemia patients (*Kharas et al., 2010*).

Because of the importance of Musashi family proteins in stem and cancer cell proliferation, we sought to identify a small molecule inhibitor of MSI1 RNA-binding activity. After screening more than 30,000 compounds, we identified four inhibitors, one of which is the intermediary metabolite oleic acid. Here, we characterize the specificity and mechanism of oleic acid inhibition and identify a novel regulatory target that enables MSI1 to act as a metabolite sensor.

## Results

### Small molecule screen to identify inhibitors of Musashi-1

To screen for small molecule inhibitors of MSI1 RNA-binding activity, we developed an in vitro assay pipeline amenable to high throughput measurements. First, we tested the ability of a purified, his6-tagged MSI1 dual RRM construct (amino acids 7–192, *Figure 1—figure supplement 1A*) to bind a fragment of a previously identified SELEX aptamer (CCCR005) (*Imai et al., 2001*) using two quantitative assays: fluorescence electrophoretic mobility shift (F–EMSA) and fluorescence polarization (FP, *Figure 1B–D*)

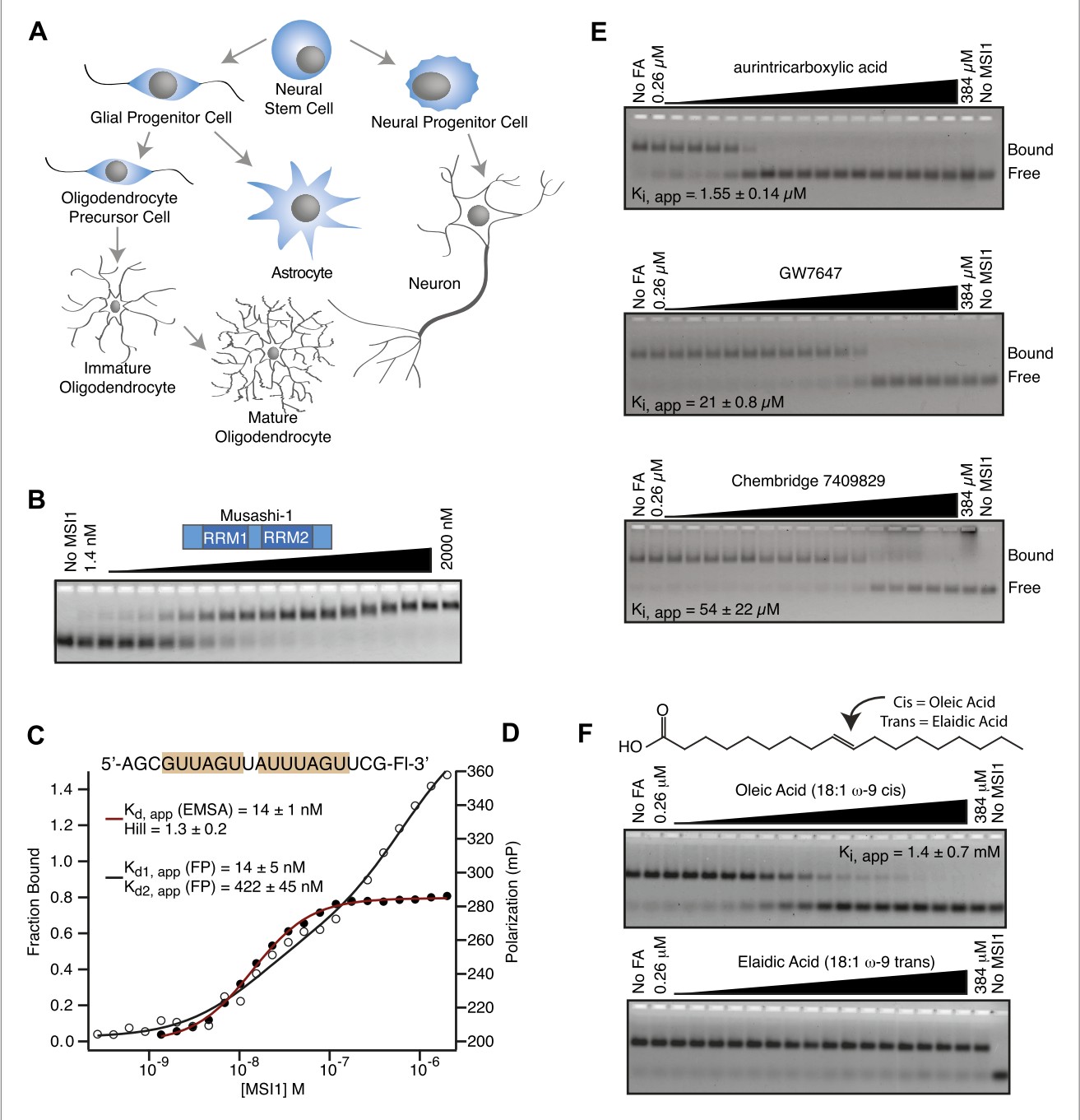

**Figure 1**. MSI1 is inhibited by monounsaturated fatty acids. (**A**) Pattern of MSI1 (blue) in the CNS. (**B–D**) EMSA and FP of MSI1 binding to RNA aptamer CCCR005 (AGCGUUAGUUAUUUAGUUCG). EMSA data (red line) were fit to the Hill equation where all shifted species were fit as an aggregate. FP data (black line) were fit to a two-site binding model. (**E** and **F**) Assay scheme for the inhibitor screen (**E**) and F–EMSA dose responses with hits identified from the small molecule screens (**E**) and oleic and elaidic acid (**F**). Each gel is one representative experiment of at least three independent experiments. No compound and no protein lanes identify the position of bound and free RNA migration, respectively.

The following figure supplements are available for figure 1:

**Figure supplement 1**.

**Figure supplement 2**. RNA binding specificity and inhibition by specific fatty acids is conserved in MSI2.

(*Pagano et al., 2011*). MSI1 binds with high affinity to the aptamer fragment, which contains two copies of the consensus sequence. Next, we optimized the FP assay for use in high throughput screens. We used this assay to screen two small molecule libraries: the 1280-compound Sigma Library of Pharmacologically Active Compounds (LOPAC) and the 30,000-compound Chembridge library (*Figure 1E*; *Table 1A*). Inhibitors identified in the screen were validated by dose response measurements using both the FP and F–EMSA assays (*Figure 1E,F*).

Four candidate inhibitors were identified. The weakest inhibitor was the Chembridge compound 7409829 ($K_{i, app, FP}$ = 15 ± 2.8 µM, $K_{i, app, F–EMSA}$ = 54 ± 22 µM). The most potent inhibitor was aurintricarboxylic acid (ATA, $K_{i, app, FP}$ = 230 ± 30 nM, $K_{i, app, F–EMSA}$ = 1.55 ± 0.14 µM), a compound that readily polymerizes in aqueous solution to form a polyanion. This compound has been identified in many high throughput small molecule assays as a non-specific inhibitor of protein–nucleic acid interactions (*Lam et al., 1995*). The next inhibitor was GW7647, a PPARα agonist (*Berger and Moller, 2002*). PPARα is a nuclear hormone receptor that is activated by long chain unsaturated fatty acids (*Gottlicher et al., 1992*; *Bocos et al., 1995*; *Forman et al., 1997*; *Kliewer et al., 1997*). The $K_{i, app}$ for this compound was 6.5 ± 0.4 µM by FP and 21 ± 0.8 µM by F–EMSA. The final inhibitor was oleic acid, an eighteen-carbon monounsaturated fatty acid with one double bond located nine carbons from the aliphatic—omega—end of the molecule (18:1, ω-9). The apparent $K_i$ of this compound was 1.2 ± 0.4 µM by FP and 1.4 ± 0.7 µM by F–EMSA, and we observed decreased affinity for the RNA aptamer with increasing concentrations of oleic acid (*Figure 1F*, *Figure 1—figure supplement 1B*). The $K_{i, app}$ is approximately two orders of magnitude below the critical micelle concentration (CMC) in our equilibration buffer (pH 8.0) as measured by N-phenyl-1-naphthylamine (CMC ≥ 75 ± 8 µM, *Figure 1—figure supplement 1C*).

Oleic acid has been screened in 754 bioassays reported in the PubChem database. Of these, oleic acid scored as positive in 4% of the assays. It should be noted that this figure overestimates the hit rate because it includes multiple bioassays that target the same protein. For example, oleic acid scored as a positive in eleven bioassays targeting fatty acid binding proteins (FABP) and seven assays that target membrane-associated potassium channels. Both are known to be sensitive to fatty acids (*Capaldi et al., 2006*; *Boland and Drzewiecki, 2008*). Other proteins responsive to oleic acid include fatty acid synthase, estrogen synthase, factor VIIa complex, and enterotoxin. It also scored positive in a screen for membrane permeant biomolecules. In total, only 14 unique proteins are responsive to oleic acid. This suggests that oleic acid inhibition is specific.

## MSI1 is specifically inhibited by 18–22 carbon ω-9 monounsaturated fatty acids

Oleic acid is the most abundant fatty acid in body fat and is produced by mature oligodendrocytes during myelination (*Martinez and Mougan, 1998*). Cells can produce almost any fatty acid by modifying existing fatty acids through precise metabolic pathways. Because oleic acid is structurally related to a large number of fatty acids, we obtained a library of fatty acids and analogs to assess the specificity of inhibition (*Figure 1F*; *Table 1*). First, we measured inhibition by longer omega-9 monounsaturated fatty acids using the FP and F–EMSA dose response assays. Eicosenoic acid (20:1, ω-9) inhibited MSI1 with a potency similar to oleic acid ($K_{i, app, FP}$ = 1.2 ± 0.4 µM, $K_{i, app, F–EMSA}$ = 1.7 ± 0.6 µM). Erucic acid (22:1, ω-9) was a stronger inhibitor ($K_{i, app, FP}$ = 640 ± 150 nM, $K_{i, app, F–EMSA}$ = 820 ± 30 nM), and nervonic acid (24:1, ω-9) inhibited more weakly ($K_{i, app, FP}$ = 47 ± 25 µM, $K_{i, app, F–EMSA}$ = 23 ± 8 µM). Second, we assessed truncations and modifications of the aliphatic end of the fatty acid. Removing two carbons (palmitoleic acid, 16:1, ω-7) had a moderate effect on inhibition ($K_{i, app, FP}$ = 5.3 ± 0.5 µM, $K_{i, app, F–EMSA}$ = 12 ± 0.9 µM). The presence of a hydroxyl group at carbon 12 (ricinoleic acid, 12-hydroxy-oleic acid) had a stronger effect; inhibition is barely detectable by FP and reduced 15-fold in F–EMSA measurements ($K_{i, app, F–EMSA}$ = 18 ± 9). Third, we assessed esterification or other modification of the carboxylate group. Oleamide, ethyl oleate, and 4-methylumbelliferyl oleate failed to inhibit RNA-binding. In contrast, the presence of a Coenzyme A (CoA) substituent was apparently tolerated, although we noted some deacylation of acyl-CoA stocks by thin layer chromatography. Fourth, we assessed the requirement for the ω-9 double bond. Stearic (18:0), palmitic (16:0), and myristic (14:0) acids failed to inhibit MSI1 RNA-binding activity, indicating that the ω-9 double bond is required. Surprisingly, the orientation of the double bond is also critical. Elaidic acid (18:1, ω-9 *trans*) has the same molecular weight as oleic acid, and nearly identical refractive index and molar aqueous solubility (~10 mM at pH 8.0), but its ω-9 double bond is *trans* rather than *cis*. Elaidic acid did not inhibit MSI1

**Table 1.** Structure-activity relationship analysis demonstrates specificity of inhibition

**A**

| Compound name | CID | Screen score | FP $K_{i, app}$ (µM) | F–EMSA $K_{i, app}$ (µM) |
|---|---|---|---|---|
| Chembridge 7409829 | 28425 | 0.045 | 15 ± 2.8 | 54 ± 22 |
| Aurintricarboxylic Acid (ATA) | 2259 | 0.053 | 0.23 ± 0.03 | 1.5 ± 0.14 |
| GW7647 | 3397731 | −0.028 | 6.5 ± 0.4 | 21 ± 0.8 |
| Oleic Acid | 445639 | −0.005 | 1.2 ± 0.4 | 1.4 ± 0.7 |

**B**

| Compound name | Structure | Code | FP $K_{i, app}$ (µM) | F–EMSA $K_{i, app}$ (µM) |
|---|---|---|---|---|
| Oleic acid |  | 18:1 ω-9 | 1.2 ± 0.4 | 1.4 ± 0.7 |
| Eicosenoic acid |  | 20:1 ω-9 | 1.2 ± 0.4 | 1.7 ± 0.6 |
| Erucic acid |  | 22:1 ω-9 | 0.64 ± 0.2 | 0.82 ± 0.03 |
| Nervonic acid |  | 24:1 ω-9 | 47 ± 30 | 23 ± 8 |
| Palmitoleic acid |  | 16:1 ω-7 | 5.3 ± 0.5 | 13 ± 0.9 |
| Linoleic acid |  | 18:2 ω-6, 9 | 2.2 ± 0.2 | 1.2 ± 0.03 |
| Arachidonic acid |  | 20:4 ω-6, 9, 12, 15 | 3.0 ± 0.2 | 1.1 ± 0.3 |
| Oleoyl-CoA |  | (18:1 ω-9) | 8.1 ± 0.3 | 4.0 ± 0.2 |
| Erucyl-CoA |  | (18:1 ω-9) | 4.1 ± 0.9 | 0.62 ± 0.2 |
| Ricinoleic acid |  | (18:1 ω-9) | No inh. | 18 ± 9 |
| Oleamide |  | (18:1 ω-9) | No inh. | No inh. |
| Ethyl oleate |  | (18:1 ω-9) | No inh. | No inh. |
| 4-Methylumbelliferyl Oleate |  | (18:1 ω-9) | No inh. | No inh. |
| Elaidic acid |  | 18:1(trans) | No inh. | No inh. |
| Stearic acid |  | 18:0 | No inh. | No inh. |
| Palmitic acid |  | 16:0 | No inh. | No inh. |
| Myristic acid |  | 14:0 | No inh. | No inh. |

(**A**) Small molecule screen hits. Compound ID (CID) refers to each compound's LOPAC identification number. Screen scores were calculated by normalizing the polarization value of each compound to the no protein and no compound controls, as described in the supplemental methods. After the screen was complete, compounds that scored as hits were confirmed by FP and F–EMSA dose response experiments. Apparent inhibition constants ($K_{i, app}$) are the average and standard deviation of at least three independent experiments. (**B**) The code = carbon number:number of double bonds, followed by the position of the double bonds from the aliphatic end of the fatty acid. Where a fatty acid is modified, the parental fatty acid numerical code is given in parentheses for comparison purposes. FP and F-EMSA dose response results are reported as the average and standard deviation of at least three independent experiments.

(*Figure 1F*). Linoleic (18:2, ω-9, ω-6) and arachidonic (20:4, ω-12, ω-9, ω-6, ω-3) polyunsaturated fatty acids also inhibit MSI1, but with a weaker apparent inhibition constant (Linoleic acid: $K_{i, app, FP}$ = 2.2 ± 0.2 μM, $K_{i, app, F–EMSA}$ = 1.2 ± 0.03 μM; Arachidonic acid: $K_{i, app, FP}$ = 3.0 ± 0.2 μM, $K_{i, app, F–EMSA}$ = 1.1 ± 0.3 μM). We conclude that omega-9 *cis* unsaturated fatty acids between 18 and 22 carbons specifically inhibit MSI1 RNA-binding activity and identify erucic acid as the most potent inhibitor (*Table 1*). MSI2 was inhibited with similar specificity (*Figure 1—figure supplement 2*).

## Oleic acid directly interacts with the RRM1 of MSI1 to effect inhibition

In principle, inhibitory fatty acids could inhibit by interacting with either MSI1 or its RNA target. We hypothesized that if the fatty acid bound to the RNA-binding domain of MSI1, the interaction might alter the local environment of its single tryptophan (W29), leading to a change in the intrinsic fluorescence (*Vivian and Callis, 2001*). To test this hypothesis, we titrated oleic acid or elaidic acid (trans-oleic acid) into recombinant MSI1 protein or an n-acetyl-L-tryptophanamide (NATA) control and measured tryptophan fluorescence. Titration of oleic acid, but not elaidic acid, strongly quenched tryptophan fluorescence and altered the emission intensity curve shape from 300 to 400 nm (*Figure 2A*, *Figure 2—figure supplement 1A–D*). Neither compound affected the fluorescence of NATA. In our buffer system, the maximal emission for both NATA and MSI1 is 350 nm (*Figure 2—figure supplement 1A–D*). We fit the emission at 350 nm to a quadratic, bimolecular association model in order to determine the apparent dissociation constant. The $K_{d, app}$ of oleic acid was 2.6 ± 1 μM, essentially identical to the $K_{i, app}$ determined by FP and F–EMSA dose response experiments (*Table 1A*; *Figure 2A*). Our data are consistent with inhibition resulting from a direct association of oleic acid with the MSI1 protein.

The NMR structures of MSI1 RRM1 and RRM2 in the absence and presence of RNA show that both domains adopt the canonical RRM fold in which an anti-parallel beta sheet is buttressed by two alpha helices (*Figure 2B*; *Nagata et al., 1999*; *Miyanoiri et al., 2003*; *Ohyama et al., 2011*). In RRM domain proteins, the beta sheet surface typically forms the RNA-binding platform (*Figure 2B*; *Kielkopf et al., 2004*; *Clery et al., 2008*). In the RNA-bound structure, several conserved amino acids located on the beta sheet surface and loops make sequence-specific contacts to the RNA (*Ohyama et al., 2011*). Notably, W29 directly contacts RNA by stacking on the first purine nucleotide. Our data reveal that fatty acid binding causes W29 fluorescence quenching, suggesting that binding changes the environment surrounding this amino acid. As such, fatty acid binding may inhibit MSI1 RNA-binding by an allosteric mechanism.

To assess whether oleic acid association induces a change in MSI1 secondary structure, we collected circular dichroism spectra as a function of fatty acid treatment. We observe a decrease in mean residue ellipticity centered around wavelength 220 nm upon treatment with oleic acid but not elaidic acid (*Figure 2C*, *Figure 2—figure supplement 1E*). To determine whether the spectral changes correspond to a change in oligomerization state, we performed velocity sedimentation analytical ultracentrifugation experiments (*Demeler et al., 2011*). We observe no significant change in the velocity sedimentation profile of MSI1 by van Holde-Weischet analysis after adding either oleic acid or elaidic acid, and the predominant species remains monomeric (*Figure 2D*) (*Demeler and van Holde, 2004*; *Demeler et al., 2011*). We conclude that oleic acid binding alters the secondary structure of MSI1, but this transition does not appear to induce aggregation at concentrations below the CMC.

After examining the published NMR structures, we were intrigued to note a hydrophobic cavity on the alpha-helical surface of RRM1, opposite the RNA-binding surface (*Figure 2B*) and adjacent to W29 (*Nagata et al., 1999*; *Miyanoiri et al., 2003*). MSI1 RRM2 does not have this feature. We hypothesized that this cavity comprises the fatty acid binding site. To test this idea, we first asked whether RRM1 is sufficient for ω-9 fatty acid inhibition. We purified his6-tagged MSI1 RRM1 (amino acids 7-103) and used FP and F–EMSA to determine whether RRM1 is sufficient for both RNA binding and inhibition by oleic acid. RRM1 bound to the aptamer RNA with an apparent $K_d$ of 75.2 ± 10 nM by FP (*Figure 2—figure supplement 2A*). Addition of oleic acid but not elaidic acid inhibits RNA-binding activity with similar $K_{i, app}$ compared to the full RNA-binding domain (*Figure 2—figure supplement 2B*). Thus, RRM1 is sufficient for RNA recognition and fatty acid inhibition.

Next, we prepared an [15]N labeled sample of RRM1 for NMR spectroscopy, collected an [15]N-[1]H HSQC spectrum, and titrated aptamer RNA to identify amide proton chemical shift differences associated with RNA binding (*Figure 2—figure supplement 2C*). The majority of the chemical shift differences map to the β-sheet surface and the loops, which is typical for RRM proteins. Notably,

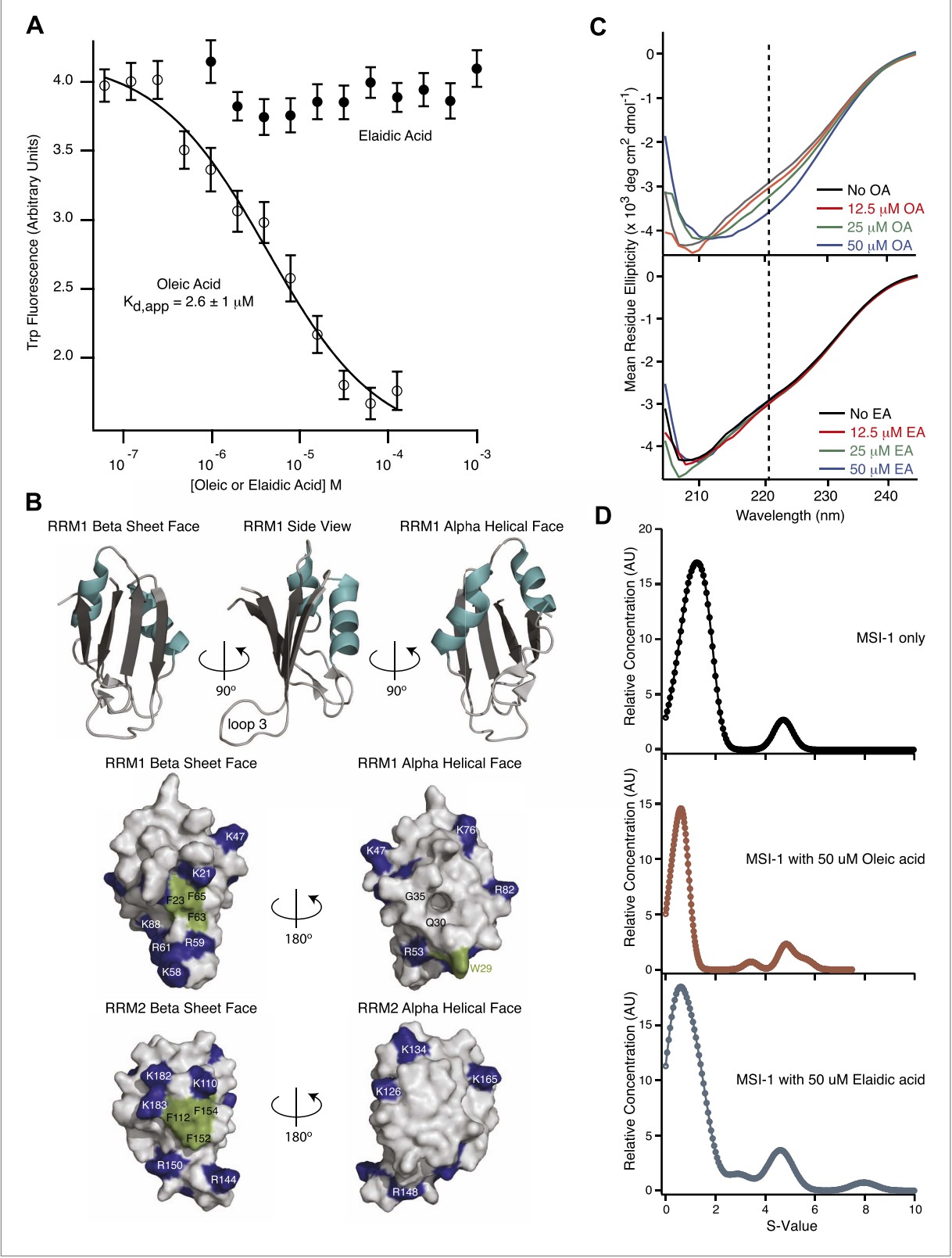

**Figure 2**. MSI1 inhibition is allosteric. (**A**) MSI1 tryptophan fluorescence at 350 nm as a function of oleic and elaidic acid. The $K_{d, app}$, $K_{i, app}$, and Hill parameters are the average and standard deviation of three independent replicates. (**B**) Ribbon model of MSI1 RRM1 (top). Space-filling model of MSI1 RRM1 (middle) and RRM2 (bottom) (***Nagata et al., 1999***; ***Miyanoiri et al., 2003***). Left, β-sheet surface, right, α-helical surface. Conserved phenylalanines

*Figure 2. Continued on next page*

*Figure 2. Continued*

and W29 are green. Lysine and arginine residues are blue. A hydrophobic pocket exists on the RRM1 α-helical surface. (**C**) CD spectra of MSI1 RRM1 in the presence of oleic (top) or elaidic acid (bottom). (**D**) Envelope traces of the van Holde-Weischet analysis for analytical ultracentrifugation experiments of MSI1 alone (top), with oleic acid (middle) and with elaidic acid (bottom). The predominant species sediments where monomeric MSI1 would be expected, and there is no significant change in the sedimentation profile after addition of oleic or elaidic acid. Data are representative traces from one of three independent experiments.

The following figure supplements are available for figure 2:

**Figure supplement 1**.

**Figure supplement 2**.

backbone amide protons corresponding to F23, G64, and F65, display large chemical shift changes upon RNA binding. In contrast, chemical shifts changes on the α helical face are small. The data are consistent with the model where the β-sheet surface recognizes RNA, and that binding involves a structural transition including F23, G64, and F65 (*Ohyama et al., 2011*). Next, we titrated oleic acid or elaidic acid into the sample of MSI (*Figure 2—figure supplement 2D–F*). We expected to observe chemical shift changes associated with oleic acid binding to the protein. The addition of increasing amounts of oleic acid did not result in significant changes in chemical shifts but in considerable reduction of the peak intensities and in line broadening. The largest loss of signal upon addition of substoichiometric concentrations of oleic acid was observed for W29, Q30, L36, C49, L50, R53, S60, G62, V74, T89, and K98 (*Figure 2*, *Figure 2—figure supplement 2E,F*). The observed loss of signal is likely due to chemical exchange between the free and bound state. As expected, we detected a strong loss in signal intensity across the entire protein when the concentration of oleic acid exceeded the CMC (75 ± 8 μM, *Figure 1—figure supplement 1C* and *Figure 2—figure supplement 2D–F*). The observed general loss of MSI1 signal is probably due to precipitation of the protein–oleic acid complex, or it could be attributed to the interaction of MSI1 with oleic acid micelles, or both. We note that a precipitant forms in the NMR tube at elevated oleic acid concentration. Because the CMC of oleic acid is below the concentration needed to make a saturated MSI1 sample (*Figure 1—figure supplement 1C*), we cannot form sufficient MSI1-oleic acid complex for NMR structural studies.

## A model of the omega-9 fatty acid inhibition mechanism

Because the high concentrations needed for NMR spectroscopic studies precluded direct measurement of the MSI1 fatty acid interface, we performed computational docking calculations with eicosenoic acid or oleic acid with MSI1 RRM1 (*Table 1*; *Figure 3*; *Friesner et al., 2004*, *2006*). In the docked models, both compounds insert the ω-9 end into the cavity, form extensive contacts with the protein surface, and position their carboxy termini adjacent to a positively charged arginine side chain (R53 or R61, dependent on the model, *Figure 3A,B*, *Figure 2—figure supplement 2D–F*, *Figure 3—figure supplement 1A*). The docked model explains the specificity of inhibition. Amide and ester derivatives lose the negative charge and cannot form charge–charge interactions with arginine (oleamide, ethyl oleate). Shorter omega-7 fatty acids will not fill the hole (palmitoleic acid). Modifications of the omega end limit insertion into the hole (ricinoleic acid). 18 carbons enable the carboxylate to reach R53 (oleic acid), (*Figure 3A,B*, *Figure 3—figure supplement 1A*). 24 carbons position the group beyond the arginine (nervonic acid). Trans orientation of the double bond limits surface contact and prevents orientation of the carboxylate towards R53 (elaidic acid). Finally, saturation of the double bond would require a large entropic penalty in order to adopt the necessary conformation (stearic acid).

To further assess how fatty acid binding alters the dynamics of amino acids that contribute to RNA recognition, we performed molecular dynamics (MD) simulations of the MSI1 RRM1 motif with and without oleic acid. The NMR structure of MS1 RRM1 served as the starting configuration for the free state (*Nagata et al., 1999*; *Miyanoiri et al., 2003*). Three different models derived by docking oleic acid provided the starting configurations for the oleic acid-bound state (*Friesner et al., 2004*, *2006*). Each of the four simulations was equilibrated for 1 ns and data were collected during a subsequent 30 ns trajectory. Upon oleic acid binding, MSI1 underwent a transition to a more open state characterized by an increase in solvent accessible surface area (SASA) and radius of gyration (*Figure 4A*). Direct visualization of the structure's time evolution showed that binding of oleic acid is associated with

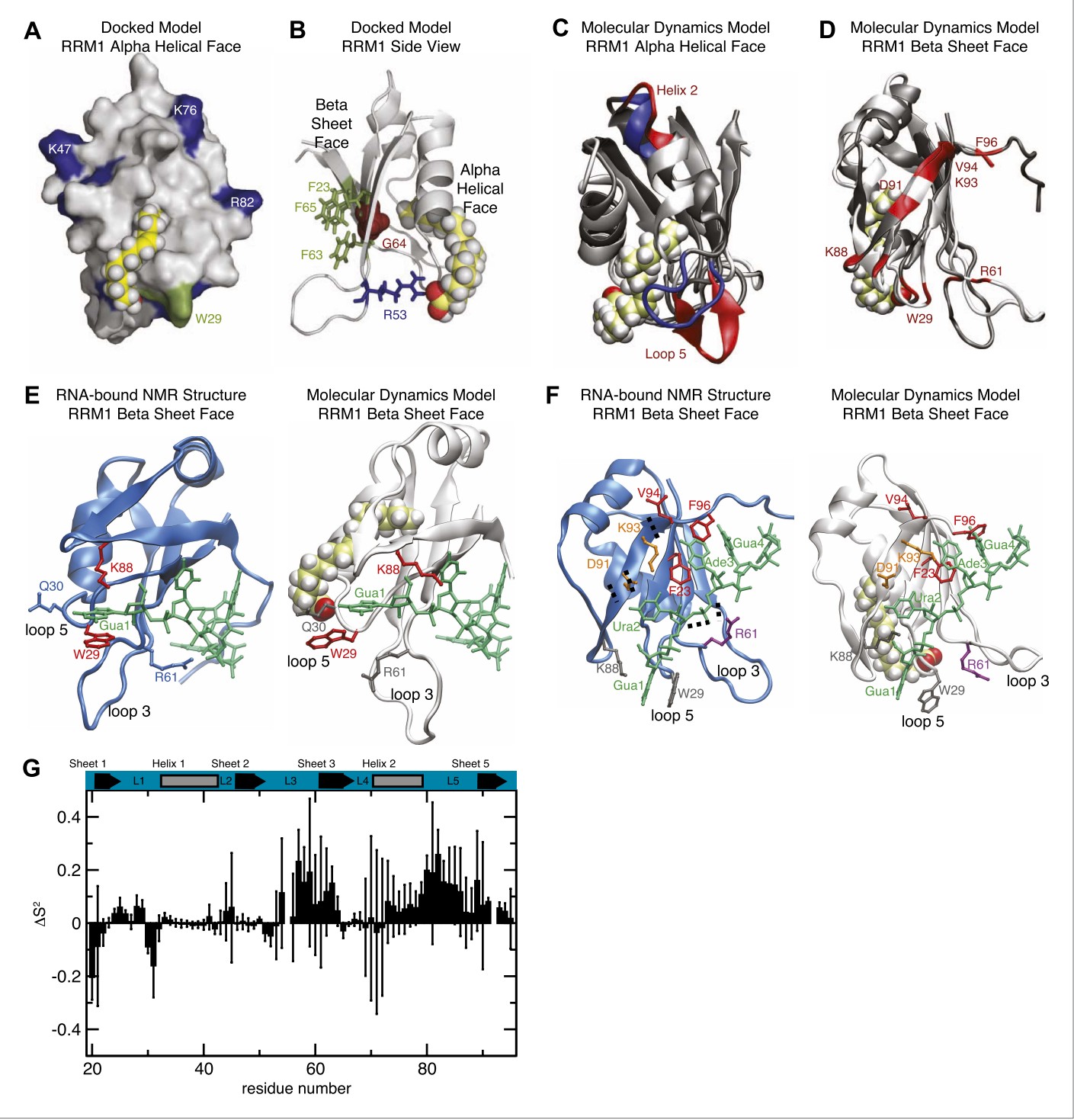

**Figure 3**. (**A** and **B**) Model of RRM1 bound to oleic acid (yellow) calculated by Schrödinger GLIDE (*Friesner et al., 2004*; *Friesner et al., 2006*). (**C**) Overlay of the oleic acid bound MD simulation (gray and red) with the apo-state NMR structure (gray and blue) (*Ohyama et al., 2011*). (**D**) RNA contact residues (red) in loop 5, helix 2, and strand 4 of the β-sheet are perturbed in oleic acid-bound molecular dynamics simulation (*Ohyama et al., 2011*). (**E** and **F**) A representative snapshot from the MD simulation of MSI1 bound to oleic acid (white) compared to the MSI1-RNA NMR structure (blue) (*Ohyama et al., 2011*). Panel (**E**) shows the Gua 1 binding pocket. In the oleic-bound state, the open conformation of loop 5 (L5) orients K88 such that K88 cannot contact Gua 1. W29 is stacked against Q30 and unavailable for stacking against Gua 1. Interaction with the side chain of R61 stabilizes the conformation of W29 in the oleic-bound state. Panel (**F**) highlights the different conformations of residues that interact with Gua 1, Ura 2, Ade 3, and Gua 4; represented in grey, orange, red, and purple, respectively. (**G**) Difference of the mean Lipari-Szabo order parameters by residue between the apo and

*Figure 3. Continued on next page*

*Figure 3. Continued*

oleic acid-bound states of MSI1. The Lipari-Szabo order parameters for the backbone NH bond vectors, $S^2$, were calculated to quantify the backbone flexibility of the free and oleic acid-bound form of MSI1. The difference of the order parameters, $\Delta S^2 = S^2_{apo} - S^2_{MSI-OA}$, indicates that MSI1-oleic acid complex is more flexible than apo MSI1, with the few exceptions mostly observed at the N-terminus. The secondary structural elements are highlighted at the top. Error bars are calculated from the standard deviation among trajectories.

The following figure supplements are available for figure 3:

**Figure supplement 1**. Mutational analysis supports molecular dynamics simulation and docked model predictions.

stabilization of the C-terminus of α-helix 1, fraying of α-helix 2 at both the N- and C-termini and, in one trajectory, formation of an additional β-sheet in loop 5 (*Figure 3C,D*; *Video 1*). The computed probability of each residue to be in a secondary structural element (*Figure 4C*) supports our observations. Notably, the β-sheet that forms in loop 5 is absent in the MSI1 apo structure but is present in the structure of MSI1 bound to RNA (*Figure 3D*; *Ohyama et al., 2011*).

Analysis of the structure and dynamics of MSI1 bound to oleic acid and its comparison to the RNA-bound structure suggests a mechanism of inhibition. Oleic acid binding stabilizes β-strand formation in loop 5 and alters its position relative to helix 1. The distance between the C-terminus of α-helix 1 and loop 5 in the free and oleic acid-bound state is greater than that observed in the RNA-bound state (*Figures 3D*, *Figure 4*). The presence of oleic acid blocks loop 5 from approaching the C-terminus of α-helix 1. When loop 5 is in the more open conformation, K88 is not in position to interact with the first purine nucleotide of the consensus, in this case Gua1. In addition, binding of oleic acid causes W29 to stack against the side chain of Q30. (*Figure 3E,F*, *Figure 4*). This observation is in agreement with the NMR signal intensity loss observed for certain residues during oleic acid titration (*Figure 2— figure supplement 2D–F*), as well as the strong quenching of MSI1 tryptophan fluorescence measured upon addition of oleic acid (*Figure 2A*, *Figure 2—figure supplement 1A–D*). W29 directly stacks with the first RNA nucleotide, stabilizing the interaction between MSI1 and RNA. Stacking is eliminated in the presence of oleic acid (*Figure 3E,F*). Finally, strand 4 is more flexible in the oleic acid-bound state than in free MSI1 (*Figure 3E,F*). Strand 4 contains several residues that directly contact RNA. The simulations suggest that oleic acid weakens these interactions. In contrast to our initial hypothesis, we do not observe a dramatic change in the dynamics of F65 in the presence of oleic acid over the time course of the simulations.

The circular dichroism and analytical ultracentrifugation data (*Figure 2C,D*, *Figure 2—figure supplement1E*) are consistent with a net gain in secondary structure upon oleic acid binding, as predicted by the MD simulation (*Figure 3*, *Figure 4*). In addition, the change in tryptophan fluorescence observed upon oleic acid but not elaidic acid treatment is consistent with the changes in tryptophan solvent exposure observed in the simulation.

To test features of the model, we mutated residues predicted to interact with oleic acid (*Figure 3— figure supplement 1*) and measured the relative inhibition constant ($K_{rel,\ app}$) by dose response FP. We observed no change in fatty acid inhibition when Q30 was mutated to a glutamate ($K_{rel,\ app} = 0.9$, *Figure 3—figure supplement 1*). Mutation of H83 to leucine also had no effect on fatty acid inhibition. Mutation of H83 to phenylalanine reduced inhibition by almost twofold. H83 lines the hydrophobic pocket. Mutation of this residue to phenylalanine is predicted to make the pocket narrower. Mutation of G64 to alanine reduced inhibition by 3.1-fold. G64 forms the floor of the hydrophobic pocket, mutation to an alanine is expected to make the pocket more shallow. Mutation of G35 to aspartate and glutamate results in a twofold and sevenfold reduction of inhibition, respectively. This is possibly due to the addition of a negative charge near the mouth of the hydrophobic cavity.

To test whether the R53 and/or R61, which are predicted to interact with the carboxylate end of oleic acid, are important for inhibition, we made single glutamate mutations at each position (R53E or R61E). Oleic acid inhibition of the R53E mutant is reduced threefold, while R61E is reduced fivefold. Mutating both arginines (R53E/R61E) increases this effect to 35-fold. This indicates that R53 and R61 are both important for inhibition, and act in a partially redundant way. A triple mutant (R53E/R59E/R61E) weakens inhibition by a similar amount (27-fold). The R53E mutant has a small effect on RNA-binding, while the R61E mutant has a strong 18-fold effect. The interaction of R61 with oleic acid would preclude its interaction with RNA, and thus may contribute to the mechanism of inhibition.

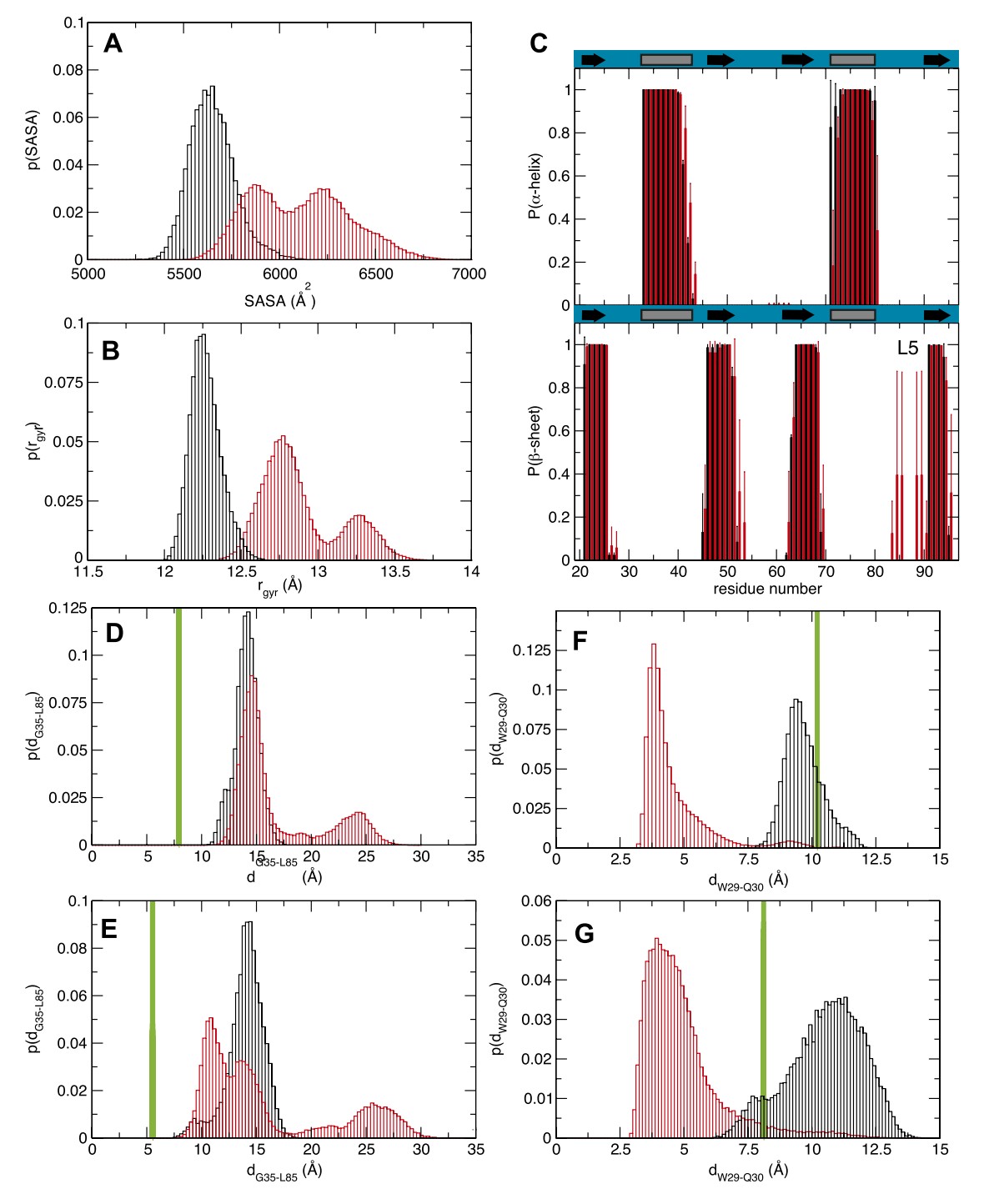

**Figure 4**. Molecular dynamics studies (**A** and **B**) represent the normalized histograms of the protein solvent accessible surface area (SASA) and radius of gyration ($r_{gyr}$) illustrating the structural transition undergone by the MSI1 upon oleic acid binding. (**A**) SASA distributions calculated from the MD trajectories of MSI1 bound to oleic acid and of MSI1 in the apo state are represented in red and black, respectively. (**B**) Radius of gyration distributions calculated from the MD trajectories of MSI-1 bound to oleic acid and of MSI1 in the apo state are represented in red and black, respectively. (**C**) The probability of being in an α-helix or β-sheet is shown for each residue of MSI1. The probabilities calculated for each residue from the MD trajectories of MSI1 free and bound to oleic acid are shown in black and red, respectively. Oleic acid binding is associated with stabilization of the C-terminus of α-helix 1, fraying of α-helix 2, at both the N- and C-termini, extension of sheet 2, as well as the formation of an additional β-sheet at loop 5 (L5). (**D** and **E**) Normalized histograms of the distance between G35, located on α-helix 1, and L85, located on loop 5, calculated from the MD trajectories of oleic acid-bound MSI-1

*Figure 4. Continued on next page*

*Figure 4. Continued*

and of apo MSI1 are shown in red and black, respectively. The distribution of distances between the $C_\alpha$ atoms of G35 and L85 is depicted in (**D**). The distribution of distances between the $C_\alpha$ of G35 and the $C_{\delta 2}$ of L85 is shown in (**E**). The green lines show the values of these distances observed in the NMR structure of MSI1 bound to RNA (PDB ID 2RS2). In the oleic-bound state, loop 5 is restricted in approaching α-helix 1 due to the steric hindrance of the oleic acid. (**F** and **G**) Normalized histograms of two representative side chain distances of W29 and Q30. Histograms calculated over the MD trajectories of oleic acid-bound MSI1 are shown in red, those of apo MSI1 in black. The distance between W29 $C_{\varepsilon 2}$ and Q30 $C_\gamma$ is shown in (**F**). The distance between W29 $C_{\zeta 2}$ and Q30 $N_{\varepsilon 2}$ is shown in (**G**). The green lines show these distances observed in the NMR structure of MSI1 bound to RNA (PDB ID 2RS2). In the oleic-bound state, the side chain of W29 is stacked against the side chain of Q30. This conformation of W29 is not observed in either the free or RNA-bound states of MSI1.

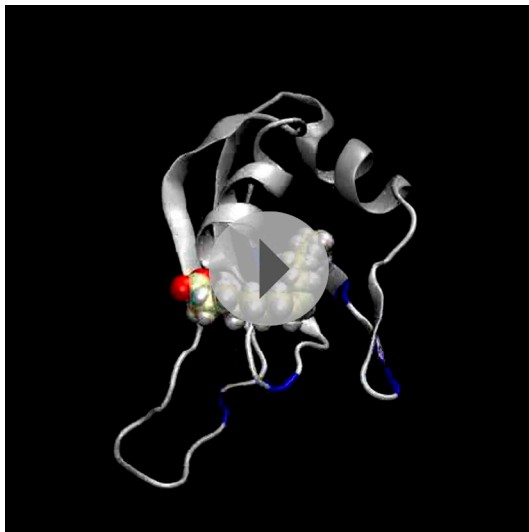

**Video 1**. This video is a representative segment of an MD trajectory of MSI1 bound to oleic acid.

Together, the MD simulations and experimental data are consistent with an allosteric model of inhibition, wherein fatty acid binding induces a change in conformation that modifies the secondary structure of RRM1 and perturbs the position of amino acids required for RNA recognition. Additional work will be necessary to determine whether fatty acid inhibition works via the mechanism suggested by the computational docking and MD simulations or through an alternative mechanism.

## Oleic acid treatment reduces oligodendrocyte progenitor cell proliferation

To study the role of fatty acids in regulating MSI1 in a cell culture model, we investigated the effect of treating the immortalized rat oligodendrocyte progenitor cell line CG-4 with oleic acid (*Louis et al., 1992*). CG-4 cells maintain normal precursor cell morphology, can be readily transfected, and can be induced to differentiate into mature oligodendrocytes by withdrawal of growth factors (*Louis et al., 1992*; *Franklin et al., 1995*). Mature oligodendrocytes produce myelin, a lipid-rich membrane structure that insulates neuronal axons to both protect them and to aid in saltatory impulse conductance throughout the CNS (*Griffiths et al., 1998*). The lipid and fatty acid profiles of mature oligodendrocyte-rich white matter differ from that of the grey matter (*Figure 5–figure supplement 3*, *Martinez and Mougan, 1998*). CG-4 cells also shift lipid profiles upon differentiation, as detected by quantitative lipidomics mass spectrometry (*Figure 5—figure supplement 1*). As with primary OPCs, MSI1 is expressed strongly in the precursor state but decreases upon differentiation (OPCs: 1.0 ± 0.3, oligodendrocytes: 0.15 ± 0.01, p-value=0.0035). Treatment of CG-4 OPCs with oleic acid strongly inhibited the rate of cell proliferation (*Figure 5*), matching the published phenotype of MSI1 knock down in primary OPCs (*Dobson et al., 2008*). In contrast, treating HEK293T cells—which do not express MSI1—with the same concentration of oleic acid had little or no effect on the rate of proliferation. Treatment with stearic acid, which does not inhibit MSI1, had no effect on proliferation rate in either cell type (*Figure 5A*). The data suggest that reduction of MSI1 activity by oleic acid limits proliferation, but we cannot rule out that the fatty acid modulates other cellular pathways that contribute to proliferation.

## MSI1 regulates stearoyl-CoA desaturase

We hypothesized that MSI1 might control the expression of enzymes required to make long chain monounsaturated fatty acids. In humans, non-dietary oleic acid is produced from stearic acid by stearoyl-CoA desaturase (SCD). Expression of SCD is tightly controlled at the transcriptional, post-transcriptional, and post-translational levels. In rodents, there are four SCD genes, *Scd1-4*. We used FP and F–EMSA assays to test the ability of recombinant MSI1 to bind each of seven putative consensus sites present in the *Scd1* 3'-UTR (*Figure 5B*). MSI1 binds to all seven sites by FP and

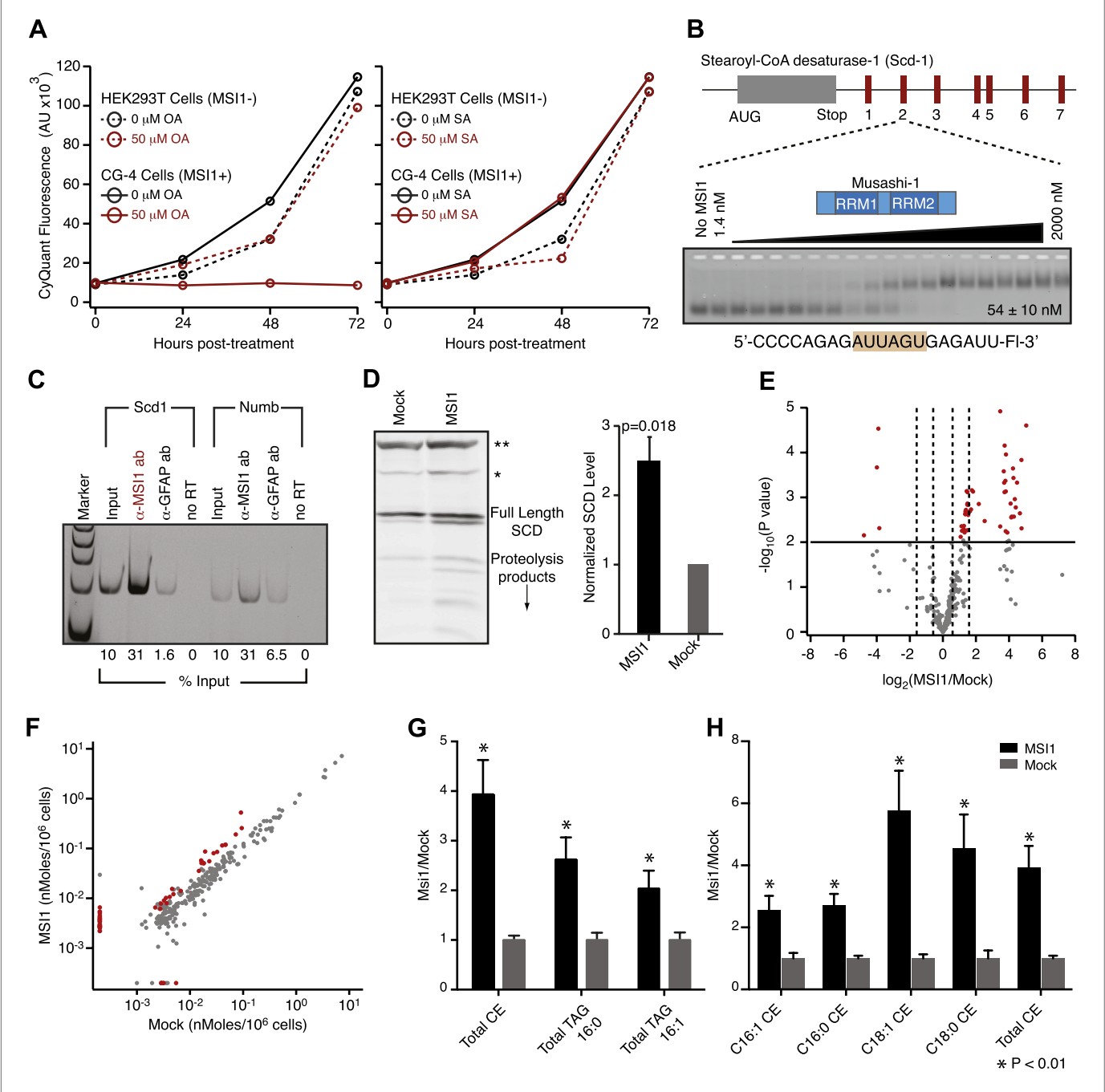

**Figure 5**. MSI-1 regulates stearoyl-CoA desaturase. (**A**) HEK293T (dashed) and CG-4 (solid) cell proliferation as a function of oleic acid or stearic acid treatment (red = treated, black = untreated). The data are the average and standard deviation of at least three biological replicates. (**B**) There are seven MSI1 consensus sites in the 3'-UTR of *Scd-1* mRNA. The $K_{d, app}$ is the average and standard deviation of at least three experiments. (**C**) *Scd-1* transcripts co-immunoprecipitate with anti-MSI1 antibodies. The data were quantified using a FUJI FLA-5000 imager. (**D**) Western analysis of SCD expression in HEK293T cells. The data were quantified using the LICOR Odyssey system relative to non specific bands (** and *, ***Figure 5—figure supplement 2B***) to control for loading. The average and standard deviation of at least three independent experiments is shown. (**E–H**) Lipidomics analysis of HEK293T cells ± MSI1 expression. Source data are included in ***Figure 5—source data 1***. (**E**) Volcano plot of lipidomics data. Dashed lines denote fold-changes of ±1.5 and ±3. Red data points indicate lipids that are significantly changed upon MSI1 expression. (**F**) Scatter plot of lipidomics data. Data are reported as nMoles per million cells. Red data points indicate lipids that are significantly changed upon MSI1 expression (FDR = 0.05). (**G**) Fold-changes of the total cholesterol esters and two TAG classes in which 38 of the 54 significantly changing lipids are categorized. Each class changes significantly with MSI1

*Figure 5. Continued on next page*

*Figure 5. Continued*

overexpression (p<0.05). (**H**) Fold-changes for the four lipids that comprise the total cholesterol esters class. All display significant changes with MSI1 expression (FDR = 0.05).

The following source data and figure supplements are available for figure 5:

**Source data 1**. Lipidomics data files.

**Figure supplement 1**. Dose response for oleic acid treatment in cell culture.

**Figure supplement 2**.

**Figure supplement 3**. Lipidomics analysis of undifferentiated and differentiated CG4 oligodendrocyte progenitor cells.

F–EMSA (*Figure 5—figure supplement 2*). Site 2 and site 7 bind to MSI1 with comparable affinity to the selected aptamer. The slightly decreased binding affinity measured for these 7 fragments is likely due to the fact that our aptamer CCCR005 contains two MSI1 binding sites, while each *Scd1* 3'-UTR fragment contains only a single MSI1 binding site. Transcripts encoding SCD1 co-immunopreciptated with MSI1 from CG-4 cell extracts using two independent antibodies (*Figure 5—figure supplement 2C*). Similar results were obtained with *Numb* mRNA, a positive control. Overexpression of MSI in HEK293T cells, which do not endogenously express MSI1, increased the amount of SCD and SCD proteolysis products by 2.5 ± 0.35-fold (p-value=0.018, *Figure 5D*, *Figure 5—figure supplement 2B*). We were unable to detect SCD expression in the rat CG-4 cells using available antibodies. qRT-PCR analysis of HEK293T mRNA shows a slight but non-significant increase in SCD1 transcript abundance upon MSI1 overexpression (*Figure 5—figure supplement 2C*).

Next, we sought to assay downstream effects of SCD regulation by MSI1 in cells. We reasoned that because SCD plays an integral role in lipogenesis, changes in SCD activity would result in a shift in the cellular lipid profile. To this end, we performed quantitative lipidomics mass spectroscopy analysis. CG-4 cell proliferation halts upon MSI1 knockdown (*Dobson et al., 2008*). For this reason, we performed these experiments in HEK293T cells with and without MSI1 expression. We observed statistically significant changes ranging from 2- to 32-fold in 54 of the 312 lipids assayed (FDR = 0.05, *Figure 5E,F*). We noted positive changes in 50 of 54 significant data points, indicating that MSI1 expression stimulates production of certain lipids. 38 of the significantly changing lipids fell into one of three cholesterol and triacylgylcerol (TAG) classes: cholesterol esters, 16:0 acyl-containing TAG, and 16:1 acyl-containing TAG (*Figure 5G,H*). Cholesterol esters and TAGs are made using SCD products, and have been shown to be more abundant in mice overexpressing SCD1, and less abundant in liver SCD1 knockout mice (*Miyazaki et al., 2000*; *Attie et al., 2002*). This indicates that SCD activity is upregulated in the presence of MSI1. Together, our binding data, co-immunoprecipitation, western blots, and lipidomics profiling results show that stearoyl-CoA desaturase expression is increased in the presence of MSI1, possibly through direct association with consensus motifs in the 3'-UTR of SCD transcripts. We note that of 2107 annotated 3' UTRs for genes involved with lipid, cholesterol, fatty acid biosynthesis, and SREBP activity, 1275 contain at least one MSI1 binding element. While the presence of a binding element does not confer regulation, MSI1 regulation of lipid metabolism may be more complex than simple direct regulation of SCD. Indeed, a survey of 64 genes that associate with MSI1 in a RIP-ChIP experiment reveals that 8 are annotated to be involved with lipid metabolic processes (*de Sousa Abreu et al., 2009*).

## Discussion

Our data reveal that the long chain ω-9 fatty acids oleic acids between 18 and 22 carbons in length are allosteric inhibitors of MSI1 RNA binding activity. Our data also show that SCD, the enzyme that catalyzes the ω-9 desaturation, is a MSI1 regulatory target. The results are consistent with a model where MSI1 controls cellular proliferation through a feedback loop that includes SCD and its enzymatic products, such as oleic acid (*Figure 6*).

Oleic acid is the precursor for synthesis of longer chain fatty acids such as eicosanoic, erucic, and nervonic acid, signaling molecules such as arachidonic acid, endocannabinoids, and prostaglandins, and membrane phospholipids. Oleic acid is abundant in the lipid-rich myelin membranes produced

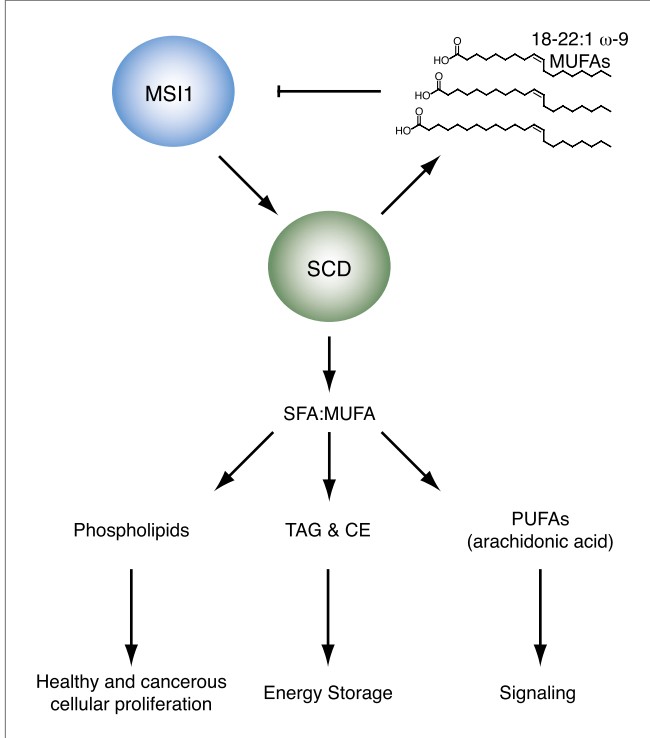

**Figure 6**. Model of SCD regulation by MSI1, and subsequent downstream consequences of SCD activity changes.

by mature oligodendrocytes (*Martinez and Mougan, 1998*), and CG-4 OPCs show global changes in the lipid profile upon differentiation. OPCs express MSI1, but mature myelinating oligodendrocytes do not (*Dobson et al., 2008*). In the oligodendrocyte lineage, oleic acid, and MSI1 levels are anti-correlated, which suggests a possible biological role for MSI1 inhibition by fatty acids (*Martinez and Mougan, 1998*; *Dobson et al., 2008*). Another MSI1 inhibitor identified in our screen was the PPARα agonist GW7647. PPARs are nuclear hormone receptor proteins that regulate cellular processes including fatty acid metabolism, differentiation, cell cycling, and inflammation (*Berger and Moller, 2002*). PPARγ agonists accelerate oligodendrocyte maturation (*De Nuccio et al., 2011*). Both GW7647 and oleic acid function as PPAR agonists, and both inhibit MSI1 RNA-binding activity, suggesting that MSI1 and PPARs possibly regulate gene expression in a reciprocal fashion.

We have identified the metabolic enzyme SCD as regulatory target of MSI1. The two human isoforms of SCD, SCD1 and SCD5, catalyze the conversion of saturated fatty acids (SFAs) into monounsaturated fatty acids (MUFAs) (*Zhang et al., 1999*; *Beiraghi et al., 2003*; *Minville-Walz et al., 2010*). MUFAs are then used in the synthesis of numerous lipids, including phospholipids, di- and triacylglycerols, cholesterol esters, and signaling molecules such as eicosanoids. SCD is therefore an essential enzyme to normal cellular proliferation, metabolism, and signaling. It is also possible that MSI1 may be regulating other factors involved in lipid, fatty acid, and cholesterol metabolism. The Penalva group identified 64 putative MSI1 targets using a RIP-ChIP analysis in HEK293T cells (*de Sousa Abreu et al., 2009*). Of these putative targets, eight are annotated to play a role in lipid, fatty acid, or cholesterol metabolic processes. These include three variants of the fatty acid elongase ELOVL5, two variants of the glycolipid biosynthetic factor PIGF, two variants of Lamin B receptor (LBR), and aminoadipate-semialdehyde dehydrogenase-phosphopantetheinyl transferase (AASDHPPT).

SCD has been implicated in a number of disease states, including obesity, diabetes, hyperlipidemia, and cancer. Obese and diabetic animals produce abnormally high levels of TAGs and cholesterol esters (*Coleman and Lee, 2004*). These energy storage molecules are also elevated in a hyperlipidemia mouse model that overexpresses SCD (*Attie et al., 2002*). Conversely, SCD1 knockout mice display impaired biosythesis of both TAGs and cholesterol esters (*Miyazaki et al., 2000*). SCD is upregulated in tumor cells, including colonic and esophageal carcinoma (*Li et al., 1994*). Cancerous cells use newly synthesized lipids primarily for phospholipid production, which are used in new membranes (*Swinnen et al., 2000*). Additionally, new evidence suggests that SCD1 activates the oncogenic Akt and AMPK signaling pathways (*Scaglia and Igal, 2008*; *Scaglia et al., 2009*).

SCD regulates SFA and MUFA homeostasis, and therefore plays an integral role in lipid signaling pathways. SFAs serve as proinflammatory factors by acting as ligands for immune receptors such as those in the Toll-like receptor family (*Shi et al., 2006*; *Nguyen et al., 2007*). MUFA products of SCD are further modified to become polyunsaturated fatty acids, such as arachidonic acid, which are converted into eicosanoids by cyclooxygenases (*James et al., 2000*). Eicosanoids serve as proinflammatory and immune signaling molecules (*Liu et al., 2011*). Recent data also link SCD to Wnt signaling through

mediation of the palmitic/palmitoleic acid conversion, as active Wnt proteins require conjugation of palmitoleic acid (*Rios-Esteves and Resh, 2013*). The balance between SFAs and MUFAs must be tightly controlled, and while SCD is not the only enzyme involved in the process, it does serve as an essential gatekeeper in the conversion of dietary SFA to MUFA.

Several examples of functional interactions between transcription factors and fatty acids have been published. For example, fatty acids associate with membrane-bound sterol responsive element binding protein (SREBP) to inhibit a cleavage event that produces the activated form of SREBP, a transcription factor involved in lipid homeostasis (*Wang et al., 1994*). Free fatty acids have been shown to stimulate proinflammatory cytokine expression while decreasing anti-inflammatory cytokine expression in adipocytes (*Bradley et al., 2008*). Interestingly, treatment of various tumor lines with oleic acid results in transcriptional inhibition of the Her-2/*neu* (*erb*B-2) oncogene through PEA3, although the precise mechanism of regulation remains unknown (*Menendez et al., 2006*). In yeast cells, fatty acids bind oleate-activated transcription factors (OAFs) to effect transcription of genes responsible for fatty acid metabolism, glucose metabolism, stress response, and other related processes (*Gurvitz and Rottensteiner, 2006*). Although numerous transcription factors are regulated by fatty acids, to our knowledge, MSI1 is the first example of a fatty acid-responsive RNA binding protein.

Integration of metabolite sensing and post-transcriptional regulation is widespread in bacteria, but relatively few examples have been found in eukaryotes (*Winkler and Breaker, 2003*; *Roth and Breaker, 2009*). Riboswitches regulate gene expression at the RNA level in response to intermediary metabolites. Small molecule metabolites bind mRNA transcripts, usually in the 5′ UTR, to induce a structural change that interferes with the transcriptional or translational machinery (*Winkler and Breaker, 2005*). In bacteria, riboswitches sensitive to a number of metabolites have been characterized, including guanine, adenine, coenzyme B12, glycine, lysine, and thyamine pyrophosphate (TPP), among others (*Grundy et al., 2003*; *Mandal et al., 2003*, *2004*; *Sudarsan et al., 2003*; *Vitreschak et al., 2003*; *Mandal and Breaker, 2004*). Although most riboswitches have been identified within the bacterial mRNA 5′ UTR, TPP riboswitches, have also been identified in plants and fungi (*Kubodera et al., 2003*; *Sudarsan et al., 2003*). While riboswitches comprise an essential cis-regulatory mechanism in bacteria, other mechanisms of coupling metabolic state to post-transcriptional gene regulation may occur in eukaryotes.

An increasing number of protein-metabolite-RNA interactions have now been identified in bacteria and eukaryotes. One example is the *Bacillus subtilis* tryptophan RNA-binding attenuation protein (TRAP). Increasing intracellular tryptophan levels induce TRAP multimerization, which enables mRNA recognition and subsequent translational repression of targets (*Gollnick et al., 1990*; *Antson et al., 1995*; *Babitzke et al., 1995*; *Yakhnin et al., 2004*). Intriguingly, several metabolic enzymes have been proposed to 'moonlight' as RNA-binding proteins in 'RNA/Enzyme/Metabolite (REM)' networks (*Ciesla, 2006*; *Hentze and Preiss, 2010*; *Castello et al., 2012*). A notable example is cytosolic aconitase, which demonstrates mutually exclusive enzymatic and RNA-binding functionality, depending upon cellular iron levels (*Hentze and Argos, 1991*; *Rouault et al., 1991*). A number of metabolite-sensitive enzymes have also demonstrated RNA-binding activity, including GAPDH, glutamate dehydrogenase, thymidylate synthase, and dihydrofolate reductase (*Ryazanov, 1985*; *Chu et al., 1991*, *1993*; *Preiss et al., 1993*; *Dollenmaier and Weitz, 2003*). This growing body of research suggests that metabolite-mediated post-transcriptional regulation is much more prevalent than previously thought.

Our results show that MSI1 N-terminal RRM1 acts as a metabolite sensor, the first example of such activity observed for the most abundant RNA-binding motif in eukaryotic genomes. A survey of RRM structures in the protein data bank reveals several with a surface cavity on the alpha helical face, which may comprise metabolite-binding pockets. We predict that a network of RNA regulatory proteins act as metabolite sensors, possibly replacing the bacterial riboswitch regulation that appears to have been largely lost in eukaryotes.

## Materials and methods

### Plasmids

DNA encoding the mouse MSI1 RNA binding domain fragment (amino acids 7–192) was amplified from the mammalian gene collection (MGC) full-length ORF clone 100014969 (Invitrogen) using gene specific primers (forward primer: 5′-cgcgcggatcccagcccggcctcgcctcccc-3′; reverse primer:

5'-gcgcgaagcttcggggacatcacctcctttg-3'). This fragment was digested using BamHI and HindIII restriction enzymes and subcloned into a modified version of pET-22b vector (Life Technologies, Grand Island, NY) in which the pelB leader sequence was replaced with a His6-Gly tag followed by a TEV protease site to make pET-22HT-MSI (7–192). Mutant versions of the MSI1 RNA binding domain were prepared by site-directed mutagenesis using QuikChange (Stratagene). The human MSI2 RNA binding domain fragment (amino acids 8–193) was amplified from MGC full-length ORF clone 3505639 using gene specific primers (forward primer: 5'-cgcgcggatccggcacctcgggcagcgccaa-3'; reverse primer: 5'-gcgcgaagctttcatgggaacatgacttctttcg-3'). This fragment was cloned into the BamHI and HindIII restriction sites of pET-22HT to make pET-22HT-MSI2 (8–193). The MSI1 RRM1 plasmid pET-22HT-MSI1 (7–103) was prepared by site-directed mutagenesis using QuikChange (Stratagene) to replace M104 with an ochre stop codon. Full-length mouse MSI1 was amplified from MGC full-length ORF clone 100014969 (Life Technologies) using forward primer 5'-cgcgcggatccatggagactgacgcgcccca-3' and reverse primer 5'-ccgggcggccgctcagtggtacccattggtgaa-3'. The resulting fragment was subcloned into pCDH-CMV-MCS-EF1-Puro (System Biosciences) using the BamHI and NotI restriction enzymes to make pCDH-CMV-MSI1(FL).

## Purification of recombinant proteins

H6-TEV-MSI-1 (7-192), H6-TEV-MSI-1 (7-104), and H6-TEV-MSI-2 (8-294) were expressed and purified from *Escherichia coli* BL21(DE3) cells. Liquid cultures grown at 37°C were induced for 3 hr during mid-log phase with 1 mM isopropyl β-D-1-thiogalactopyranoside (IPTG). Cells were pelleted, resuspended in lysis buffer (50 mM $NaH_2PO_4$, 300 mM NaCl, 20 mM Imidazole, 5 mM β-Mercaptoethanol [BME]), and lysed using a microfluidizer (IDEX Health and Science). Soluble lysate was applied to a Ni-NTA column (Qiagen), washed with wash buffer (50 mM $NaH_2PO_4$, 300 mM NaCl, 50 mM Imidazole, 5 mM BME), and eluted with elution buffer (50 mM $NaH_2PO_4$, 300 mM NaCl, 300 mM Imidazole, 5 mM BME). Fractions were analyzed by SDS-page and those containing recombinant MSI1 or MSI2 were pooled and dialyzed overnight into S buffer (50 mM MOPS pH 6.0, 20 mM NaCl, 2 mM DTT). Pooled fractions were applied to a HiTrap SP cation exchange column (GE Healthcare) and eluted using a gradient of 0.1 M–1 M NaCl in S Buffer. Fractions containing MSI1 or MSI2 were pooled and dialyzed overnight into Q buffer (50 mM Tris pH 8.8, 20 mM NaCl, 2 mM DTT) prior to loading a HiTrap Q anion exchange column (GE Healthcare). The protein was eluted over a gradient from 0.1 to 1 M NaCl in Q buffer. Fractions containing MSI were pooled and dialyzed using Spectra/Por 7 25 kD (MSI1) or 10 kD (MSI2) molecular weight cutoff tubing (Spectrum laboratories) overnight into storage buffer (50 mM Tris pH 8.0, 20 mM NaCl, 2 mM DTT). The yield of >95% pure MSI1 or MSI2 is typically 20 mg per liter of culture (*Figure 1—figure supplement 1*).

## RNA sequences and labeling

Synthetic RNA oligonucleotides were ordered from IDT and 3' end-labeled with fluorescein 5-thiosemicarbazide (Life Technologies) according to the method of Reines and Cantor (*Reines and Cantor, 1974*; *Pagano et al., 2007*; *Farley et al., 2008*). Briefly, $5 \times 10^{-10}$ mol of RNA were incubated with 100 mM NaOAc, pH 5.1, and 5 nmol of $NaIO_4$ for 90 min at room temperature then ethanol precipitated. The RNA was resuspended in 50 μl of 1 mM fluorescein-5-thiosemicarbazide in 100 mM NaOAc, pH 5.1. After incubating overnight at 4°C, the RNA was separated from unreacted label by ethanol precipitation and subsequently passaged through a Pierce centrifuge column packed with Spehadex G-25 resin (GE Healthcare). RNA CCCR005 was a truncated form of the SELEX aptamer S8-13 identified by the Okano lab (*Imai et al., 2001*). In preparation for the small molecule screen, fCCCR005 (AGCGUUAGUUAUUUAGUUCG/36-FAM/) was ordered pre-labeled from IDT.

## Fluorescence polarization and electrophoretic mobility shift assays

Fluorescence polarization (FP), also known as fluorescence anisotropy, and Fluorescence electrophoretic mobility shift assays (F–EMSA) were used to measure the binding affinity of recombinant MSI1 to fluorescein-labeled RNA aptamers. Assays were conducted as described in *Pagano et al. (2011)*. Briefly, 2 nM fluorescein-labeled RNA was incubated with varying concentrations of recombinant purified MSI1 protein in equilibration buffer (37.5 mM Tris pH 8.0, 75 mM NaCl, 0.0075% igepal, 0.0075 mg/ml tRNA) for 3 hr. Fluorescence polarization was determined with a Victor V3 plate reader using a 480 ± 31 nm excitation filter and a 535 ± 40 nm emission filter. After measuring FP, the samples were mixed with 6 ×x bromocresol green loading dye (0.15% (wt/vol) Bromocresol green, 30% (vol/vol) glycerol) and run on a 5% native polyacrylamide gel at 120 V for 75 min at 4°C. Wet gels were

scanned with a Typhoon FLA 9000 Biomolecular imager (GE healthcare) using a 473 nm laser and a long-pass cut-off filter (510 nm). The fraction of bound RNA was determined by quantifying lower (free) and upper (bound) band intensities using MultiGauge and ImageGauge software (Fujifilm). For RNA sequences with two shifted species, the bands were quantified together.

Polarization values or the fraction of bound RNA were plotted as a function of protein concentration and fit to the Hill *Equation 1* to determine the apparent dissociation constant ($K_d$) and the apparent Hill coefficient ($n$). The upper ($m$) and lower ($b$) values were also fit in order to define the assay window.

$$\Phi = b + (m-b)\left[\frac{1}{1+\left(K_d/[P_t]\right)^n}\right] \tag{1}$$

For RNA sequences where a bi-phasic transition was observed, the FP data were fit using a two-site model (*Equation 2*) to determine both apparent dissociation constants ($K_{d1}$ and $K_{d2}$) and the fraction ($F$) of signal that corresponds to each transition.

$$\Phi = \left[F(m-b)\left(\frac{P_t}{P_t - K_{d1}}\right)\right] + \left[(1-F)(m-b)\left(\frac{P_t}{P_t + K_{d2}}\right)\right] + b \tag{2}$$

## Small molecule screen

The small molecule screen was performed at the UMass Medical School small molecule screening core facility using a variation of the FP assay described above. 100 nM recombinant MSI1 protein and 2 nM fluorescein-labeled RNA aptamer CCCR005 were added to each assay well of 384-well black plates (Corning) using a µFill liquid dispenser (BioTek). Compounds dissolved in DMSO from the LOPAC and Chembridge libraries of small molecules were spotted in each assay well to a final concentration of 384 µM using a Tecan genesis workstation 150. 64 wells of each plate were reserved for controls, including 32 wells that included no protein and no compound (free RNA), and 32 wells that included protein, RNA, and DMSO (no compound). The plates were equilibrated at 25°C prior to collecting polarization and fluorescence intensity data for each well using a Victor V2V plate reader (Perkin Elmer). The $Z'$ score, a measure of signal-to-noise, was calculated for each plate using the average ($\mu$) and standard deviation ($\sigma$) of the control wells (*Equation 3*).

$$Z^1 = 1 - \left[\frac{3\left(\sigma_1 + \sigma_2\right)}{\mu_1 - \mu_2}\right] \tag{3}$$

Plate reads with a Z' of <0.5 were repeated. The average Z' of all plates was 0.7 ± 0.2. The polarization values ($mP$) of each well were normalized against the assay window using the mean polarization values of the no protein control ($\mu_2$) and no compound control ($\mu_1$) wells to generate an assay score (*Equation 4*).

$$Score = \frac{mP - \mu_2}{\mu_1 - \mu_2} \tag{4}$$

Hits were classified as wells with a score of 0.1 or less where the fluorescence intensity remained within twofold of the control average to eliminate false positives due to compound fluorescence or quenching.

## Dose response experiments

Dose response experiments to assess inhibition activity were performed using a modified FP and F–EMSA protocol. A constant concentration of sub-saturating protein was equilibrated with 2 nM fluorescein-labeled RNA and varying concentrations of compound in equilibration buffer. The FP and F–EMSA data were collected as above and fit to a sigmoidal dose response equation to determine the IC50 (*Equation 5*).

$$\Phi = b + (m-b)\left[\frac{1}{1+\left(IC_{50}/[P_t]\right)^n}\right] \tag{5}$$

The apparent inhibition constant was calculated using the Lin and Riggs conversion (*Equation 6*), which corrects for the equilibrium dissociation constant of MSI1 for the labeled RNA as well as the

concentration of labeled RNA and protein used in the experiment (*Lin and Riggs, 1972*; *Ryder and Williamson, 2004*).

$$K_{i,app} = \frac{2(K_d)(IC_{50})}{2P - R - 2K_d}$$ (6)

## Intrinsic tryptophan fluorescence assay

To directly assay the association of fatty acids and MSI1 protein, 6 µM MSI1 or N-acetyl-tryptophanamide (NATA) was incubated with varying concentrations of compound. Equilibrated reactions were excited at 280 nm, then steady-state fluorescence emission spectra were recorded between 295 and 400 nm in 1 nm intervals using a T-format Fluorolog fluorimeter (Horiba). 350 nm emission data were normalized and fit to a quadratic bimolecular association curve (*Equation 7*) to determine the apparent dissociation constant, where $C$ is the total compound concentration, $P$ is the total protein concentration, and $m$ and $b$ represent the maximal and minimal signal, respectively.

$$\Phi = b + (m - b)\left[\frac{C + P + K_d - \sqrt{(C + P + K_d) - 4(CP)}}{2C}\right]$$ (7)

## In-silico docking analysis

The ligand molecules were downloaded from the PubChem database (pubchem.ncbi.nlm.nih.gov) and prepared for docking using the LigPrep module in Maestro (Schrödinger, LLC). The target protein structures were downloaded from the PDB (www.rcsb.org) and prepared for docking in Maestro (Schrödinger, LLC) using the Protein preparation wizard. Glide (Schrödinger, LLC) was used to generate the receptor grid for subsequent docking and scoring the docked ligands in Standard Precision (SP) mode. The pose with best Glide score from each ligand/receptor docking run was selected for further analysis (*Friesner et al., 2004*; *Halgren et al., 2004*; *Friesner et al., 2006*).

## NMR

Labeling with $^{15}$N was performed by growing cells in isotopically enriched M9 medium, 1 g $^{15}$NH$_4$Cl per liter. 2D $^1$H-$^{15}$N HSQC spectra were collected using samples of U-$^{15}$N MSI1 in 90% H$_2$O/10%D$_2$O buffer solution of 50 mM Tris at pH 7.0. 2D $^1$H-$^{15}$N HSQC spectra were collected for each incremental addition of the unlabeled ligand (either aptamer RNA, oleic acid, or elaidic acid) to the $^{15}$N MSI1 sample to determine amide proton chemical shift changes upon titration of the ligand. All experiments were performed at 600 MHz on a Varian Inova spectrometer equipped with a triple-resonance cold probe at 298 K. Data processing was performed using NMRPipe (*Delaglio et al., 1995*) and sparky (Goddard and Kneller) software.

## Molecular dynamics simulation

We performed molecular dynamics (MD) simulations of the RRM1 domain of MSI (residues 20–96) free and bound to oleic acid. We modeled the unknown structure of MSI1 bound to oleic acid by starting from the NMR solution structure of MSI1 RRM1 (*Miyanoiri et al., 2003*; *Ohyama et al., 2011*) and docking the oleic acid ligand using the GLIDE software package from Schrödinger, LLC (*Friesner et al., 2006*), followed by energy minimization and equilibration. All structures were solvated and neutralized in a TIP3P water box (*Jorgensen et al., 1986*). Energy minimization and MD simulations were subsequently carried out using the NAMD software package (*Phillips et al., 2005*) and using the version 27 CHARMM potential energy function (*MacKerell et al., 1998*). The particle mesh Ewald method (*Darden et al., 1993*; *Essmann et al., 1995*) was used to treat electrostatic interactions and periodic boundary conditions were applied throughout. The SHAKE algorithm (*Ryckaert et al., 1977*) was applied throughout the simulation to constrain the hydrogen atom bond lengths at their equilibrium values and an integration time step of 2 fs was used. After an initial energy minimization, the system was simulated in the isothermal-isobaric ensemble. Non-bonded interactions were calculated every time step using a cut-off distance of 12 Å. After equilibration in the isothermal-isobaric ensemble, an additional stage of equilibration was performed in the microcanonical ensemble. We then collected three independent 30 ns constant-NVE production runs of MSI-1 bound to oleic acid and one for MSI-1 in the free-state at an average temperature of 298 K.

Visualization and secondary structure analysis was performed in VMD, using the STRIDE method (*Frishman and Argos, 1995*).

In addition, we employed a number of measures to characterize the structural and dynamical changes between the free and bound state of MSI-1, including the radius of gyration, the solvent accessible surface area (SASA), distance between backbone and/or side chain atoms of residues G35-L85 and W29-Q30, and Lipari and Szabo order parameters (*Lipari and Szabo, 1982*; *Humphrey et al., 1996*). These quantities were calculated using VMD (*Humphrey et al., 1996*) software along with bespoke programs previously described elsewhere (*Morgan and Massi, 2010*).

## Circular dichroism

Far-UV circular dichroism (CD) spectra were collected using 10 µM MSI1 in 50 mM Tris pH 8.0, 100 mM NaCl, and 0.1% TFE on a Jasco-810 spectropolarimeter (Jasco Inc., Easton, MD). Spectra were collected from 215 to 260 nm in a 0.2-cm path length quartz cuvette using a scan rate of 20 nm min$^{-1}$ and a response time of 8 s. The sample temperature for all CD measurements was maintained at 293 K.

## Analytical ultracentrifugation

Sedimentation velocity analyses were conducted using a Beckman Optima XL-I analytical ultracentrifuge in the University of Massachusetts Medical School Ultracentrifuge Facility. Data were analyzed with UltraScan III version 1.0 (*Demeler et al., 2011*). Sedimentation velocity experiments were performed with 45 µM MSI1 in storage buffer (50 mM Tris-HCl pH 8.0, 20 mM NaCl, 2 mM DTT). Measurements were made at 20°C using an AN60ti rotor at 20,000 rpm and 280 nm in intensity mode. Partial specific volumes were estimated on the basis of peptide sequence with UltraScan and found to be 0.7280 cm$^3$/g for the MSI1 RBD. Data were analyzed by two-dimensional spectrum analysis with simultaneous removal of time and radially invariant noise (*Demeler et al., 2009*; *Brookes et al., 2010*). Noise and diffusion-corrected, model-independent sedimentation coefficient distributions were generated using the enhanced van Holde–Weischet analysis (*Demeler et al., 1997*; *Demeler and van Holde, 2004*).

## Cell-based experiments

### Cell lines and culture

CG-4 rat oligodendrocyte progenitor and B104 neuroblastoma cells were a gift from Lynn Hudson and were cultured as previously described (*Louis et al., 1992*). CG-4 cells were maintained in the progenitor state in 30% B104 conditioned media. HEK293T cells were cultured in Dulbecco's modified Eagle's medium (Life Technologies) with 10% fetal bovine serum (Atlanta Biologicals Inc., Lawrenceville, GA), 100 units/ml penicillin, and 100 µg/ml streptomycin (Invitrogen).

### Proliferation assays

100,000 HEK293T or CG-4 cells were seeded into each well of six-well culture plates. 24 hr after seeding, cells were treated with either 50 µM oleic acid in EtOH or EtOH only as a control. 24, 48, and 72 hr after fatty acid treatment, cells were assayed for proliferative changes using the Cyquant direct cell proliferation assay (Life Technologies) according to the manufacturer's instructions using a Victor V3 plate reader (PerkinElmer). The treatment concentration was chosen after a dose response for toxicity and efficacy across a range defined by established protocols for neuroblastoma cells (*Figure 5—figure supplement 1*; *Di Loreto et al., 2007*).

### RNA co-immunoprecipitation

For each experiment, 10$^6$ CG-4 cells were crosslinked for 10 min with 0.1% formaldehyde in PBS then scraped into lysis buffer (1% [wt/vol] SDS, 10 mM EDTA, 1 ×x EDTA-free protease inhibitors). Lysate was diluted in 10X IP buffer (0.01% SDS, 1.1% Triton X-100, 1.2 mM EDTA, 16.7 mM Tris–HCl, pH 8.1, 167 mM NaCl) with 8 units SUPERase-In (Ambion). Immunoprecipitation was performed using rabbit polyclonal anti-Musashi 1 (ab21628; Abcam) or rabbit polyclonal anti-GFAP (ab7260; Abcam) conjugated to Dynabeads protein G (Life Technologies) according to the manufacturer's instructions. After proteinase K digestion (NEB) and heat-induced crosslink reversal, RNA was phenol–chloroform extracted, ethanol precipitated, and resuspended in 20 µl nuclease-free water. RNA was then treated with DNase (Ambion Turbo DNAfree) to remove any DNA contamination. RNA was reverse transcribed and amplified using a SuperScript III One-Step RT-PCR System with PlatinumTaq (Life Technologies) and transcript-specific fluorescein-labeled primers for *scd1* and *numb*. PCR products were separated from free primers on a 5% native polyacrylamide gel, imaged on a Fuji FLA-5000 imager, and quantified using Image Gauge software.

## MSI1 overexpression and qRT-PCR

24 hr prior to transfection, $10^6$ HEK293T cells were seeded in 10-cm culture plates. The cells were transfected with 2 µg pCDH-CMV-MSI1(FL) or empty pCDH-CMV-MCS-EF1-Puro using Effectine reagent according the manufacturer's instructions (Qiagen). 24 hr after transfection, transfected cells were selected for with 2 µg/ml puromycin (Sigma). RNA and protein were harvested 72 hr after transfection. To confirm MSI1 overexpression, RNA was collected using Trizol reagent (Life Technologies) followed by DNAse treatment (Turbo DNAfree, Ambion) according to the provided instructions. RNA yield was determined by spectrophotometry and quantitative RT-PCR was performed with an Opticon thermal cycler (Bio-Rad) using an iScript One-Step RT-PCR Kit with SYBR Green (Bio-Rad) according to the manufacturer's instructions. All assays were performed in triplicates. Data were analyzed by sigmoidal curve fitting according to the method of *Rutledge (2004)* and normalized to GTF2i or Tubulin.

## SCD western blots

Cells were lysed with SDS lysis buffer (1% SDS, 10 mM EDTA, 1x EDTA-free protease inhibitors) and protein concentration was determined by the bicinchoninic acid assay method (BCA, Pierce) according to the manufacturer protocol. Samples were boiled for 5 min after the addition of 6 ×x Laemmli buffer (9% SDS and 60% glycerol, 375 mM Tris-HCl pH 6.8, 0.015% Bromophenol blue, 12% β-mercaptoethanol). Proteins were separated for 2 hr on a 12% SDS-polyacrylamide gel using a Bio-Rad mini-PROTEAN electrophoresis apparatus, transferred to low-fluorescence PVDF membranes (Millipore) for 2 hr at 4°C at a constant 70 V in 25 mM Tris, 150 mM glycine, and 10% (vol/vol) methanol transfer buffer, blocked with Odyssey Blocking Buffer (OBB, LI-COR Biosciences) for 1 hr at room temperature, and probed with 1:1000 dilution of mouse monoclonal anti-SCD antibody (ab19862; Abcam) in OBB overnight at 4°C. After washing with PBS-T, membranes were incubated with fluorescent goat anti-mouse secondary antibodies (926-32210; LI-COR Biosciences) at 1:15000 in OBB for 1 hr at room temperature. Signal was detected using an Odyssey Infrared Imaging System (LI-COR Biosciences). To ensure antibody specificity, shRNA knockdown of SCD1 was performed with two independent shRNA constructs (V3LHS_305870 and V3LHS_305872; Open Biosystems) and compared with non-silencing shRNA construct (RHS4346; Open Biosystems).

## Total lipid extraction

Cells were removed from plates with 0.25% Trypsin-EDTA (Gibco) and the total cell number was obtained using a hemocytometer. Each sample contained $1–2 × x10^6$ cells. After pelleting by centrifugation, cells were resuspended in 200 µl $H_20$, immediately followed by 250 µl methanol and 125 µl chloroform/0.01% Butylated hydroxytoluene (BHT). Samples were shaken vigorously for 30 s, then an additional 250 µl chloroform/0.01% BHT and 250 µl $H_20$ was added. After 30 s of additional shaking, samples were centrifuged for 5 min at 4°C. The chloroform layer was removed to a new tube, and the cell sample was treated with 250 µl of chloroform/0.01% BHT, vigorous shaking, centrifuging and subsequent chloroform layer removal twice more. Combined chloroform layers were washed with 500 µl KCL, then 500 µl $H_2O$. Insoluble layers were transferred to glass vials and dried under argon. Samples were sent to the Kansas lipidomics research center for analysis by mass spectrometry as previously described (*Welti et al., 2002*). Five independent biological replicates were analyzed for each treatment group.

## Lipidomics data analysis

The lipid profile data were acquired at Kansas Lipidomics Research Center (KLRC). Instrument acquisition and method development at KLRC was supported by NSF grants MCB 0455318, MCB 0920663, DBI 0521587, DBI 1228622, Kansas INBRE (NIH Grant P20 RR16475 from the INBRE program of the National Center for Research Resources), NSF EPSCoR grant EPS-0236913, Kansas Technology Enterprise Corporation, and Kansas State University. Lipidomics data were reported as nMoles/$1 × 10^6$ cells. After removing data points for samples with values below the limit of detection for the instrumentation (0.02 nMoles/$1 × 10^6$ cells), all data points for which no signal was detected in either treatment group were removed. A pseudocount of 0.002 nMoles/$1 × 10^6$ cells was added to all remaining data points. Outliers were eliminated using a Q test. Significant differences between treatments for each lipid class were determined using a Student's *t* test and an FDR of 0.05.

## Critical micelle concentration determination

The critical micelle concentration (CMC) for oleic acid in equilibration buffer was determined using N-phenyl-1-naphthylamine (NPN), a compound that fluoresces when sequestered into micelles (*Hagihara*

*et al., 2002*; *Hasegawa et al., 2008*). 25 µM NPN was incubated with varying amounts of oleic acid for 30 min. Fluorescence intensity was determined with a Victor V3 plate reader using a 355 nm excitation filter and a 460 nm emission filter. Fluorescence intensity data were plotted as a function of fatty acid concentration. The CMC was determined by a two state segmented linear regression to identify the breakpoint between baseline fluorescence and fluorescence caused by NPN association with micelles.

## Acknowledgements

The authors thank Dr Ruth Zearfoss, Dr Kendall Knight, and Dr Phillip Zamore for helpful discussions, Dr Hong Cao for technical assistance with the small molecule screen, Dr Jill Zitzewitz for helpful discussions and advice concerning CD measurements, Dr Osman Bilsel and Sagar Kathuria for technical support and advice with intrinsic tryptophan fluorescence measurements, Sarah Swygert, Kim Crowly, Jeremy Mann, and Dr Borries Demeler for help and advice with velocity sedimentation experiments, Alicia Bicknell and Dr Melissa Moore for help and sharing equipment and reagents for IR western blot experiments, Dr C Robert Matthews for advice and sharing equipment, and the team at the Kansas lipidomics research center for quantitative lipidomics analysis and advice on preparing samples. This work was supported by NIH Grant GM098763 to FM and NIH Grants GM081422 and GM098643 to SPR.

## Additional information

### Funding

| Funder | Grant reference number | Author |
| --- | --- | --- |
| National Institutes of Health (NIH) | GM081422 | Sean P Ryder |
| National Institutes of Health (NIH) | GM098643 | Sean P Ryder |
| National Institutes of Health (NIH) | GM098763 | Francesca Massi |

The funders had no role in study design, data collection and interpretation, or the decision to submit the work for publication.

### Author contributions

CCC, LMD, FM, Conception and design, Acquisition of data, Analysis and interpretation of data, Drafting or revising the article; SAH, Acquisition of data, Analysis and interpretation of data, Drafting or revising the article; RMG, SMDS, Acquisition of data, Analysis and interpretation of data; SPR, Conception and design, Analysis and interpretation of data, Drafting or revising the article

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
