## [Decision Letter]

Thank you for sending your work entitled “Allosteric inhibition of a stem cell RNA-binding protein by an intermediary metabolite” for consideration at *eLife*. Your article has been favorably evaluated by James Manley (Senior editor) and 3 reviewers, one of whom is a member of our Board of Reviewing Editors.

The following individuals responsible for the peer review of your submission have agreed to reveal their identity: Douglas Black (Reviewing editor).

The Reviewing editor and the other reviewers discussed their comments before we reached this decision, and the Reviewing editor has assembled the following comments to help you prepare a revised submission.

The authors identify an allosteric inhibitor of an RNA binding protein that allows inducible control of gene expression at the RNA level. This protein, Musashi, is an important regulator of translation during development and had not previously been suspected to have such properties. The authors show inhibition of RNA binding by a specific subset of fatty acids (18-22 carbon ω-9 monounsaturated fatty acids). They propose that the lipids bind to the first of two RNA recognition motif (RRM) domains in Musashi-1. They further identify stearoyl-CoA desaturase (SCD), which produces non-dietary oleic acid, as an mRNA target of Musashi-1, indicating a feedback regulatory mechanism. This is perhaps the first example of RNA binding protein regulation by a small molecule ligand, similar to the many examples of transcriptional regulators, such as the nuclear hormone receptors, that are regulated by small ligands. The biochemistry is carefully done. However, many of the interpretations rest on structural modeling and some initial lipidomics in cell culture, and there are several experiments that would solidify their conclusions.

Major Points:

1) The proposed binding site for oleic acid on Musashi RRM1 is supported almost exclusively from model building – in part because the concentration of fatty acid needed to solve the structure of the bound complex by NMR was not achievable. It would strengthen their conclusions considerably if they could test some of the predictions from modeling by mutagenesis. Can they eliminate the binding of oleic acid with precise point mutations? Would these mutants exhibit constitutive RNA binding in vitro and in cells? There are 2 possible arginines on the surface (R53 and R61) that are candidates for interaction with the fatty acid. Also, RRM2 lacks the fatty acid binding hole, so what fills the space? Could that information be used to engineer RRM1, so that it won't bind to oleic acid?

2) The authors introduce the CG-4 cells as a physiologically relevant context for MSI regulation. But they then primarily use the HEK system. Is there a technical reason for not using CG-4 cells? This would allow knockdown of MSI and assessment of oleic acid response, changes in SCD1 expression and changes in lipids in the absence of MSI.

3) The lipidomic data show extensive changes in lipid metabolism in response to MSI1. It seems likely that some of these may be due to direct or indirect effects on other regulators of fatty acid and cholesterol synthesis, in addition to SCD1. The authors should at least comment on this. For example, are there changes in SREBP proteins?

---

## [Author Response]

*1) The proposed binding site for oleic acid on Musashi RRM1 is supported almost exclusively from model building – in part because the concentration of fatty acid needed to solve the structure of the bound complex by NMR was not achievable*. *It would strengthen their conclusions considerably if they could test some of the predictions from modeling by mutagenesis. Can they eliminate the binding of oleic acid with precise point mutations? Would these mutants exhibit constitutive RNA binding in vitro and in cells? There are 2 possible arginines on the surface (R53 and R61) that are candidates for interaction with the fatty acid. Also, RRM2 lacks the fatty acid binding hole, so what fills the space? Could that information be used to engineer RRM1, so that it won't bind to oleic acid?*

We cloned, expressed, and purified 10 mutant versions of MSI1 to test various residues implicated in the oleic acid interaction in our MD simulation and the docked model. The results of the suggested experiments are consistent with our previous hypothesis and strengthen the manuscript considerably. They are presented in the Results section of the manuscript and in Figure 4–figure supplement 2. In short, mutations that make the hydrophobic pocket smaller weaken fatty acid inhibition. Mutation of a glycine at the top of the channel to a charged amino acid also reduces fatty acid inhibition. Mutation of R53 or R61 reduces fatty acid inhibition as well, and mutation of both at the same time has a large effect on inhibition. These data support our model, in which the hydrophobic cavity and one of two basic arginine residues on the flexible loop comprise the fatty acid binding site.

*2) The authors introduce the CG-4 cells as a physiologically relevant context for MSI regulation. But they then primarily use the HEK system. Is there a technical reason for not using CG-4 cells? This would allow knockdown of MSI and assessment of oleic acid response, changes in SCD1 expression and changes in lipids in the absence of MSI*.

There are two primary technical reasons that prohibit use of CG-4 cells in our

SCD assays. First, it has been shown that MSI1 knockdown halts proliferation and induces apoptosis in primary oligodendrocyte progenitor cells (28). We have confirmed this result in rat CG-4 oligodendrocyte progenitor cells. Second, we attempted to blot for SCD from CG-4 cells, but were unable to detect SCD protein using commercially available antibodies. This is likely because CG-4 cells are a rat cell line, and there is no robust SCD antibody available that targets rat SCD. We have edited the manuscript accordingly to clarify our rationale for using HEK cells in these experiments.

*3) The lipidomic data show extensive changes in lipid metabolism in response to MSI1. It seems likely that some of these may be due to direct or indirect effects on other regulators of fatty acid and cholesterol synthesis, in addition to SCD1*. *The authors should at least comment on this. For example, are there changes in SREBP proteins?*

This is a good point, and we agree that indirect effects on lipid metabolism via deregulation of other MSI1 targets are a distinct possibility. To address this, we looked at all genes annotated to be involved with lipid, cholesterol, fatty acid biosynthesis, and SREBP activity. This list includes genes involved in biosynthesis, metabolism, positive and negative regulation, or simply known lipid/fatty acid binding factors. There were 2107 such genes with annotated 3’ UTRs in the human genome. Of these, 1275 contained at least one MSI1 binding site in its 3’ UTR. Eight of these have been shown to associate with MSI1 in published RIP-CHIP data sets (21).

We have added these results to the manuscript. We have also modified our discussion to include the possibility of indirect regulation and/or more MSI1 targets in these pathways.